# OUTLIER-ROBUST ORTHOGONAL REGRESSION ON MANIFOLDS

## ABSTRACT

Motivated by machine learning and computer vision applications, we formulate
the problem of Outlier-Robust Orthogonal Regression to find a point in a manifold
that satisfies as many linear equations as possible. Existing approaches addressing
special cases of our formulation either lack theoretical support, are computation-
ally costly, or somewhat ignore the manifold constraint; the latter two limit them
from many applications. In this paper, we propose a unified approach based on
solving a non-convex and non-smooth $\ell^1$ optimization problem over the manifold.
We give conditions on the geometry of the input data, the manifold, and their inter-
play, under which the minimizers recover the ground truth; notably the conditions
can hold even when the inliers are skewed within the true hyperplane. We pro-
vide a Riemannian subgradient method and an iteratively reweighted least squares
method, suiting different computational oracles, and prove their linear/sub-linear
convergence to minimizers/critical points. Experiments demonstrate that respect-
ing the manifold constraints increases robustness against outliers in robust essen-
tial matrix estimation and robust rotation search.

## 1 INTRODUCTION

Given a dataset $\mathcal{Y} = [\boldsymbol{y}_1, \ldots, \boldsymbol{y}_L] \in \mathbb{R}^{D \times L}$ of $L$ points and a manifold $\mathcal{M} \subset \mathbb{R}^D$, we propose
*Outlier-Robust Orthogonal Regression* (OR$^2$), formulated using the $\ell^0$ semi-norm as

$$\min_{\boldsymbol{b}} \quad \left\| \mathcal{Y}^\top \boldsymbol{b} \right\|_0 \quad \text{s.t.} \quad \boldsymbol{b} \in \mathcal{M}, \quad \boldsymbol{b} \neq \boldsymbol{0}. \qquad \text{(OR}^2\text{-}\ell^0\text{)}$$

In words, (OR$^2$-$\ell^0$) is an $\ell^0$ *regression* problem of finding some vector $\boldsymbol{b}^*$ from $\mathcal{M}$ that is *orthogonal*
to as many points in $\mathcal{Y}$ as possible. We assume such $\boldsymbol{b}^*$ is orthogonal to $N$ points in $\mathcal{Y}$ ($N \leq L$).
Points of $\mathcal{Y}$ orthogonal to $\boldsymbol{b}^*$ are called *inliers*, or otherwise *outliers*. By solving (OR$^2$-$\ell^0$) one not
only finds $\boldsymbol{b}^*$, but also identifies inliers and outliers, thus (OR$^2$-$\ell^0$) is an *outlier-robust* formulation.

OR$^2$ has massive applications in *machine learning* such as robust subspace recovery ($\mathcal{M}$ is the unit
sphere), hyperplane clustering (the Chow variety), fixed-rank matrix sensing (fixed-rank matrices),
and in *geometric vision* including robust rotation search ($\mathbb{SO}(3)$), essential matrix estimation (the
essential manifold), and trifocal tensor estimation (the calibrated trifocal tensor variety), among
many others. To further motivate the paper, we begin with a tour of two important instances of OR$^2$.

**OR$^2$ over the Sphere $\mathbb{S}^{D-1}$.** In the late 1980s, Spath & Watson (1987) first considered the problem

$$\min_{\boldsymbol{b}} \quad \left\| \mathcal{Y}^\top \boldsymbol{b} \right\|_1 \quad \text{s.t.} \quad \boldsymbol{b} \in \mathcal{M} = \mathbb{S}^{D-1}, \qquad (1)$$

and called it $\ell^1$ *orthogonal regression*, which motivates the name of this paper. (1) can be viewed
as the $\ell^1$ relaxation of (OR$^2$-$\ell^0$) with a choice of $\mathcal{M} = \mathbb{S}^{D-1}$ to fix $\boldsymbol{b}$'s scale, since (OR$^2$-$\ell^0$) is
invariant to scalings. It was not until the past decade that a fruitful line of research considered (1)
(sometimes with other surrogate objectives) in robust subspace recovery (Tsakiris & Vidal, 2015;
2017; 2018b; Zhu et al., 2018; Ding et al., 2019a; 2021), dictionary learning (Qu et al., 2014; Sun
et al., 2017b;a; Bai et al., 2018; Gilboa et al., 2018), and sparse blind deconvolution (Qu et al.,
2019); see Qu et al. (2020) for a summary. Despite being non-convex non-smooth, (1) enjoys strong
guarantees of successful recovery. For example, Zhu et al. (2018) shows that (1) is robust to as many
outliers as the square of the number of inliers, and Zhu et al. (2018) furthermore provides linearly
convergent algorithms to find $\boldsymbol{b}^*$.

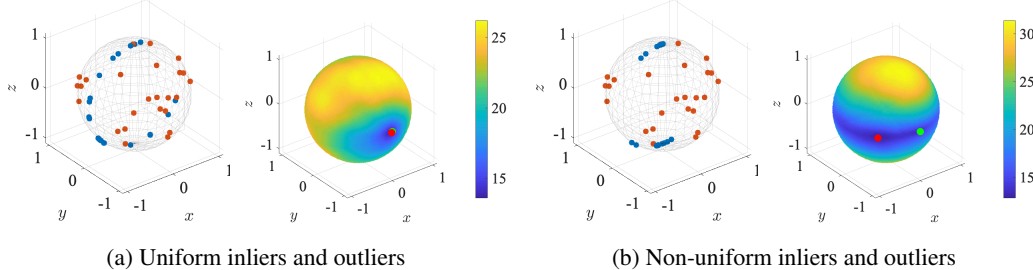

(a) Uniform inliers and outliers        (b) Non-uniform inliers and outliers

Figure 1: Inliers (blue) and outliers (orange) over $\mathbb{S}^2$ (left); objective of (1) over $\mathbb{S}^2$ (right), with the true $\boldsymbol{b}^*$ (green) and the minimizer (red).

**$OR^2$ over the Essential Manifold.** Given two images of a static scene taken from a camera that undergoes a rotation $\boldsymbol{R}^* \in \mathbb{SO}(3)$ and translation $\boldsymbol{t}^*$, how can we estimate $\boldsymbol{R}^*, \boldsymbol{t}^*$? This is a classic textbook question (Ma et al., 2003; Hartley & Zisserman, 2004) in geometric vision and has been a building block of modern autonomous driving systems. To solve it, some matching algorithm is typically run first on the two images to produce *correspondences* $\{(\boldsymbol{z}_j, \boldsymbol{z}_j')\}_{j=1}^L \subset \mathbb{S}^2 \times \mathbb{S}^2$. *Inlier correspondences* $(\boldsymbol{z}_j, \boldsymbol{z}_j')$ satisfy

$$\boldsymbol{z}_j^\top \boldsymbol{E}^* \boldsymbol{z}_j' = 0 \quad \Leftrightarrow \quad \mathrm{tr}(\boldsymbol{z}_j' \boldsymbol{z}_j^\top \boldsymbol{E}^*) = 0, \tag{2}$$

where $\boldsymbol{E}^* = [\boldsymbol{t}^*]_\times \boldsymbol{R}^*$ is the so-called *essential matrix*[1], lying in the *essential manifold*

$$\mathcal{M}_E := \{\boldsymbol{E} \in \mathbb{R}^{3\times3} : \quad \boldsymbol{E} = [\boldsymbol{t}]_\times \boldsymbol{R}/\sqrt{2}, \quad \boldsymbol{t} \in \mathbb{S}^2, \quad \boldsymbol{R} \in \mathbb{SO}(3)\}, \tag{3}$$

in which we restricted $\boldsymbol{t} \in \mathbb{S}^2$ as (2) is invariant to the scale of $\boldsymbol{t}$. Note then that (2) is equivalent to

$$\langle \boldsymbol{y}_j, \boldsymbol{b}^* \rangle = 0, \quad \text{where} \quad \boldsymbol{y}_j = \mathrm{vec}(\boldsymbol{z}_j' \boldsymbol{z}_j^\top), \quad \boldsymbol{b}^* = \mathrm{vec}(\boldsymbol{E}^*), \tag{4}$$

which naturally leads to two different approaches for estimating $\boldsymbol{E}^*$ (and therefore $\boldsymbol{R}^*$ and $\boldsymbol{t}^*$):

• (*Essential Relaxation*?) Observe that $\boldsymbol{E}^*$ satisfies $\mathrm{tr}(\boldsymbol{E}^*(\boldsymbol{E}^*)^\top) = 1$, so $\boldsymbol{b}^* = \mathrm{vec}(\boldsymbol{E}^*)$ lies in a *proper* subset of $\mathbb{S}^8$. Therefore a first approach is to relax the constraint $\boldsymbol{E}^* \in \mathcal{M}_E$ and simply assume that $\boldsymbol{b}^*$ in (4) lies in the unit sphere $\mathbb{S}^8$. This idea is classic in geometric vision and has received significant popularity (Hartley & Zisserman, 2004). In particular, with this relaxation, robustly estimating an essential matrix reduces to (1), which can be solved via diverse sets of methods mentioned above and notably (Ding et al., 2020b), with powerful theoretical guarantees. However, these guarantees all build on the assumption that inliers and outliers are somewhat uniformly distributed, which is violated here. Specifically, the entries of each $\boldsymbol{y}_j = \mathrm{vec}(\boldsymbol{z}_j' \boldsymbol{z}_j^\top)$ in (4) are correlated; it has been further proved that often $\boldsymbol{y}_j$'s are *clustered* in some proper subspace (Ding et al., 2020b) under certain *degeneracies*, rather than uniformly distributed. Figure 1 presents a visual analogy in $\mathbb{R}^3$. On the left of Figure 1a (*resp.* Figure 1b), the blue points are uniform (*resp.* non-uniform) inliers and orange points are outliers. Then we see that, for uniform inliers, (1) has a unique up-to-sign minimizer in red coinciding with $\boldsymbol{b}^*$ in green (Figure 1a, right), while non-uniform inliers, in contrast, lead to an arbitrary minimizer far from $\boldsymbol{b}^*$ (Figure 1b, right).

• (*$OR^2$ on $\mathcal{M}_E$*?) To address the issue, one can consider $\ell^1$ orthogonal regression directly on $\mathcal{M}_E$:

$$\min_{\boldsymbol{b}} \quad \left\| \mathcal{Y}^\top \boldsymbol{b} \right\|_1 \quad \text{s.t.} \quad \boldsymbol{b} \in \mathcal{M} = \mathcal{M}_E. \tag{5}$$

That being said, we are not aware of any existing theoretical guarantees for (5) regarding its minimizers or recovery algorithms. More broadly, to the best of our knowledge, such guarantees have rarely if ever been derived for many other similar outlier-robust estimation problems in geometric vision that admit an orthogonal regression formulation and are associated naturally with manifold constraints (e.g., rotations). This thus invokes a strong call for carrying out a theoretical and principled investigation into orthogonal regression in a general, manifold optimization context.

---

[1]For any $\boldsymbol{a} \in \mathbb{R}^3$, $[\boldsymbol{a}]_\times := \begin{bmatrix} 0 & -a_3 & a_2 \\ a_3 & 0 & -a_1 \\ -a_2 & a_1 & 0 \end{bmatrix}$ is a matrix representing cross product, i.e., $[\boldsymbol{a}]_\times \boldsymbol{b} = \boldsymbol{a} \times \boldsymbol{b}$.

**Contributions.** Going beyond unit spheres and essential manifolds, we propose to study

$$\min_{\boldsymbol{b}} \quad \left\| \mathcal{Y}^\top \boldsymbol{b} \right\|_1 \quad \text{s.t.} \quad \boldsymbol{b} \in \mathcal{M} \qquad\qquad (\text{OR}^2\text{-}\ell^1)$$

where $\mathcal{M}$ is assumed to be a smooth and compact submanifold of the unit sphere $\mathbb{S}^{D-1}$ (Assumption 1). This is a non-convex and non-smooth optimization problem. We consider two efficient algorithms for ($\text{OR}^2$-$\ell^1$): a Riemannian subgradient method (Algorithm 1) when one can compute a retraction map of $\mathcal{M}$ and a projection map onto $T_{\boldsymbol{b}}\mathcal{M}$, or otherwise an iteratively reweighted least squares method (Algorithm 2) when a weighted least squares problem over $\mathcal{M}$ can be easily solved.

• *Landscape*: We provide conditions on the geometry of the inliers, outliers, $\mathcal{M}$, and their interplay, under which the minimizer of ($\text{OR}^2$-$\ell^1$) recovers (Theorems 1 and 2) or is close to (Theorem 3) the ground truth. Notably, ($\text{OR}^2$-$\ell^1$) admits a successful recovery even if the inliers are skewed within the true hyperplane, as long as $\mathcal{M}$ is in some sense posed close to the mode of inliers (Figure 2) and does not curve too much near the ground truth (Figure 3); a novel insight to our knowledge.
• *Convergence*: We prove that the Riemannian subgradient method converges linearly to the ground truth with suitable initialization and step sizes (Theorem 4). Moreover, we show that the iteratively reweighted least squares converges globally sub-linearly to a critical point of a surrogate to ($\text{OR}^2$-$\ell^1$) (Theorem 5); this has not been shown in the literature even for particular instances of ($\text{OR}^2$-$\ell^1$).
• *Experiments*: We show that respecting the manifold constraints (as opposed to relaxing them to the unit sphere ones) increases robustness against outliers in tasks of robust estimation of essential matrix (§5) and rotation (A.2) with different parameters (e.g., number of points, outlier ratios).

## 2 PRIOR ART: OUTLIER-ROBUST ESTIMATION

Here we review two types of popular approaches for outlier-robust estimation in geometric vision.

**RANSAC.** One of the most widely applied approach for ($\text{OR}^2$-$\ell^0$) is RANdom SAmple Consensus (Fischler & Bolles, 1981) that arised in the early 1980s. The most basic version operates by iteratively i) sampling say $d < D$ points from $\mathcal{Y}$, ii) fitting to them a model from $\mathcal{M}$, iii) computing a score that measures how many points in $\mathcal{Y}$ are consensus with this model. It terminates and returns the model with the highest score, when the probability of having sampled an outlier-free model is above a prescribed threshold.

A well-known drawback of RANSAC is, however, that the iteration complexity increases dramatically with the percent of outliers in $\mathcal{Y}$ as well as $d$. Efforts mitigating the latter have unfortunately led to a dilemma: While it is possible to reduce $d$ to the theoretical lower bound, the dimension of $\mathcal{M}$, by utilizing as many algebraic constraints from $\mathcal{M}$ as possible (Nistér, 2004), yet decreasing $d$ undermines the accuracy of step ii) against noise in $\mathcal{Y}$.

We conclude by remarking that numerous heuristics have been proposed to improve RANSAC in aspects of sampling (Chum & Matas, 2005; Torr et al., 2002; Barath et al., 2019a), score function (Torr & Zisserman, 2000; Stewart, 1995; Moisan et al., 2016), degeneracy handling (Chum et al., 2005; Ivashechkin et al., 2021), refitting (Chum et al., 2003; Lebeda et al., 2012; Barath & Matas, 2018; Barath et al., 2020; 2019b; Yang et al., 2021), and combinations thereof (Raguram et al., 2013; Ivashechkin et al.). Despite their empirical success on specific problem and dataset instances, little has been done on the theoretical understanding of when and why these variants offer accurate or efficient recovery. Indeed, the only theoretical result we are aware of is on the *basic* RANSAC algorithm in the problem of robust subspace recovery (Maunu & Lerman, 2019, §4).

**Continuous Robust Surrogates.** Since the $\ell^0$ objective of ($\text{OR}^2$-$\ell^0$) is discrete, one naturally considers its continuous surrogates to faciliate continuous optimization. Indeed, people have replaced the objective in ($\text{OR}^2$-$\ell^0$) with $\ell^1$, Huber, Logcosh or truncated quadratic, albeit these works are restricted to particular instances of $\mathcal{M}$, to be reviewed in the sequel. Observe that the relaxed problem is still *non-convex* since the feasible set $\mathcal{M}$ is so.

When $\mathcal{M}$ is the set of rotations $\mathbb{SO}(3)$, Lajoie et al. (2018); Yang & Carlone (2019; 2022); Peng et al. (2022a) considers a surrogate of ($\text{OR}^2$-$\ell^0$) using the truncated quadratic objective. Notably, they established its equivalence to a quadratically constrained quadratic programming (QCQP), which is in turn relaxed to a semi-definite programming (SDP), a *convex* problem. This allows computing and analyzing the minimizer of QCQP and SDP problems under certain conditions; it

has been shown to be rather robust to noise and outliers. However, this approach scales poorly with the number of samples $L$: indeed, the SDP has $O(L^2)$ variables, hindering real-time computation for even $L = 50$.

A fruitful series of research consider instead optimizing directly some surrogates of ($OR^2$-$\ell^0$), allowing much faster computation. When $\mathcal{M}$ is $\mathbb{S}^{D-1}$, the minimizer of ($OR^2$-$\ell^1$) recovers the ground truth (Zhu et al., 2018), and (Projected) Riemannian sub-gradient method with suitable initialization and step sizes converges (piece-wise (Zhu et al., 2018)) linearly (Zhu et al., 2019; Li et al., 2019) to the minimizer. Nevertheless, these results requires, among other conditions, that the inliers to be uniformly distributed in the hyperplane orthogonal to the true $\boldsymbol{b}^*$, a condition that can often be violated, e.g., in geometric vision (see Example 1 and experiments). On the other hand, when $\mathcal{M}$ is the normalize essential variety, a family of surrogates *empirically* enjoyes accurate and efficient recovery (Zhao, 2022); this in principle applies to a rich family of $\mathcal{M}$ as long as weighted least squares problems can be solved (§3.2). Be that as it may, there is no theoretical guarantee whatsoever telling when the minimizer is the ground truth or why algorithms should converge to it.

## 3 FORMULATION

In this section, we first formulate the problem in a precise manner (§3.1), and then discuss two algorithms useful when different mappings can be efficiently computed with respect to the manifold $\mathcal{M}$ (§3.2). The theoretical analysis of the objective and algorithms are deferred to §4.

### 3.1 SET UP: DATA MODEL, ASSUMPTIONS, NOTATIONS

To begin with, we define the manifold and the data model, making clear the associated assumptions. The first assumption we make is the following.

**Assumption 1** (Regularity on manifold). $\mathcal{M}$ is a smooth and compact submanifold of $\mathbb{S}^{D-1}$.

Let $\mathcal{Y} = [\boldsymbol{y}_1, \ldots, \boldsymbol{y}_L] \in \mathbb{R}^{D \times L}$ be a given *normalized* dataset, i.e., $\boldsymbol{y}_i \in \mathbb{S}^{D-1}$. When the context is clear we also treat $\mathcal{Y}$ as the set $\{\boldsymbol{y}_i\}_{i=1}^L$ with some abuse of notation. Let $\mathcal{X} = [\boldsymbol{x}_1, \ldots, \boldsymbol{x}_N]$ be the subset of $\mathcal{Y}$ containing points from $\mathcal{Y}$ that are orthogonal to a ground truth vector $\boldsymbol{b}^* \in \mathcal{M}$, which we refer to as *inliers*. It turns out to be handy to define the ground truth hyperplane $\mathcal{H} := \operatorname{span}(\boldsymbol{b}^*)^\perp$. On the other hand, denote the points from $\mathcal{Y} \backslash \mathcal{X}$ by $\mathcal{O} = [\boldsymbol{o}_1, \ldots, \boldsymbol{o}_M]$, termed *outliers*; observe that $L = N + M$. Finally, we make a mild assumption on the outliers.

**Assumption 2** (Outliers are in general position). Any size-$D$ subset of $\mathcal{O}$ are linearly independent.

*Remark* 1. The requirements that points $\boldsymbol{y}_i$'s from $\mathcal{Y}$ lie on the unit sphere $\mathbb{S}^{D-1}$ and $\mathcal{M}$ is a submanifold of $\mathbb{S}^{D-1}$ do not compromise generality. Note that the objective of ($OR^2$-$\ell^0$) is invariant to the scale of $\boldsymbol{b}$ and points from $\mathcal{Y}$, i.e., ($OR^2$-$\ell^0$) concerns $\boldsymbol{b}$ and $\boldsymbol{y}_i$'s only in *directions* but not *scaling*. As such, for an arbitrary dataset we can project[2] its points onto $\mathbb{S}^{D-1}$ to form a qualified $\mathcal{Y}$; for an *arbitrary* smooth manifold $\overline{\mathcal{M}}$, taking $\boldsymbol{\Pi}_{\mathbb{S}^{D-1}} (\overline{\mathcal{M}} \backslash \{\boldsymbol{0}\})$ gives a $\mathcal{M}$ obeying Assumption 1.

*Remark* 2. Assumption 2 is not a strong assumption, since for outliers from a uniform distribution over $\mathbb{S}^{D-1}$ it holds with probability 1.

**Notations.** We denote the objective function of ($OR^2$-$\ell^1$) as $f(\cdot)$, i.e., $f(\boldsymbol{b}) = \left\| \mathcal{Y}^\top \boldsymbol{b} \right\|_1$. For a set $\mathcal{S} \subset \mathbb{R}^D$ and a vector $\boldsymbol{x} \in \mathbb{R}^D$, $\boldsymbol{\Pi}_{\mathcal{S}}(\boldsymbol{x}) := \operatorname{argmin}_{\boldsymbol{y}} \{ \| \boldsymbol{x} - \boldsymbol{y} \|_2^2 : \boldsymbol{y} \in \mathcal{S} \}$ is the projection of $\boldsymbol{x}$ onto $\mathcal{S}$ provided it exists. For any $\boldsymbol{v} \in \mathbb{R}^D$, $\langle \boldsymbol{v} \rangle$ denotes the linear subspace spanned by $\boldsymbol{v}$. For any linear subspace $\mathcal{S}$ of $\mathbb{R}^D$, $\mathcal{S}^\perp$ is its orthogonal complement. Let $\operatorname{sign}(\cdot)$ be such that $\operatorname{sign}(x)$ is 1 if $x > 0$, $-1$ if $x < 0$ and 0 otherwise. Denote by $T_{\boldsymbol{b}}\mathcal{M}$ the tangent space of $\mathcal{M}$ at $\boldsymbol{b} \in \mathcal{M}$. Given $\boldsymbol{b} \in \mathcal{M}$ we denote by $\operatorname{retr}_{\boldsymbol{b}} : T_{\boldsymbol{b}}\mathcal{M} \to \mathcal{M}$ the retraction map that sends a vector $\boldsymbol{v}$ in the tangent space $T_{\boldsymbol{b}}\mathcal{M}$ back to the manifold $\mathcal{M}$. We use rgrad for the Riemannian gradient.

---

[2] A minor detail here is that there is not a unique projection of the zero vector $\boldsymbol{0}$ onto $\mathbb{S}^{D-1}$, a point that we can remove from $\mathcal{Y}$ regardless without affecting minimizers of ($OR^2$-$\ell^0$).

| **Algorithm 1:** RSGM for ($OR^2$-$\ell^1$) | **Algorithm 2:** IRLS for ($OR^2$-$\ell^1$) |
|---|---|

**Algorithm 1: RSGM for ($OR^2$-$\ell^1$)**

1 Initialization $\boldsymbol{b}_0 \in \mathcal{M}$, step sizes: $\{\alpha_k > 0\}_{k=1,2,\dots}$

2 For $k \leftarrow 1, 2, \dots, K$:

$$\boldsymbol{g}_k \in \partial f(\boldsymbol{b}_k) \qquad (6)$$

$$\boldsymbol{r}_k \leftarrow \boldsymbol{\Pi}_{T_{\boldsymbol{b}_k}\mathcal{M}}\, \boldsymbol{g}_k \qquad (7)$$

$$\boldsymbol{b}_{k+1} \leftarrow \mathrm{retr}_{\boldsymbol{b}_k}(-\alpha_k \boldsymbol{r}_k) \qquad (8)$$

**Algorithm 2: IRLS for ($OR^2$-$\ell^1$)**

1 Initialization $\boldsymbol{b}_0 \in \mathcal{M}$, parameter $\delta > 0$

2 For $k \leftarrow 0, \dots, K$:

$$w_j^{k+1} \leftarrow \frac{1}{\max\{|\boldsymbol{y}_j^\top \boldsymbol{b}_k|, \delta\}}, \quad \forall j = 1, \dots, L \qquad (9)$$

$$\boldsymbol{b}_{k+1} \in \underset{\boldsymbol{b} \in \mathcal{M}}{\operatorname{argmin}} \sum_{j=1}^{L} w_j^{k+1} \cdot \left(\boldsymbol{y}_j^\top \boldsymbol{b}\right)^2 \qquad (10)$$

### 3.2 ALGORITHMS

To solve ($OR^2$-$\ell^1$), we describe two algorithms: a Riemannian subgradient method when a retraction map of $\mathcal{M}$ and a projection map onto $T_{\boldsymbol{b}}\mathcal{M}$ are accessible, or otherwise an iteratively reweighted least squares method when a weighted least squares problem over $\mathcal{M}$ can be easily solved.

**Riemannian Sub-gradient Method (RSGM).** Algorithm 1 gives the classic RSGM algorithm for ($OR^2$-$\ell^1$), which iteratively computes a subgradient (6), projects it onto the tangent space (7), and takes a retraction step (8). To give a concrete example, suppose for the moment that $\mathcal{M}$ is the unit sphere. In this case the computation of the subgradient in step (6) can be contextualized as $\boldsymbol{g}_k \leftarrow \sum_{j=1}^{L} \mathrm{sign}(\boldsymbol{y}_j^\top \boldsymbol{b}_k)\boldsymbol{y}_j$, the projection step (7) is given as $\boldsymbol{r}_k \leftarrow \boldsymbol{g}_k - \boldsymbol{b}_k \boldsymbol{b}_k^\top \boldsymbol{g}_k$, and the retraction step (8) can be implemented simply as a projection onto the sphere, $\boldsymbol{b}_{k+1} \leftarrow (\boldsymbol{b}_k - \alpha_{k+1}\boldsymbol{r}_k)/\|\boldsymbol{b}_k - \alpha_{k+1}\boldsymbol{r}_k\|_2$. In general, the subgradient step (6) can be computed for typical objective functions and the two Riemannian optimization steps (7) and (8) can be computed efficiently for a wide range of manifolds $\mathcal{M}$'s. Therefore, we assume that Algorithm 1 can be implemented efficiently, and will analyze its convergence rate in §4.3 and performance in applications in §5.

**Iteratively Reweighted Least Squares (IRLS).** Algorithm 2 presents the classic IRLS framework to solve ($OR^2$-$\ell^1$). IRLS alternates between updating the weights (9) and solving a weighted least-squares problem (10). The choice of weight $w_j^{k+1}$ in (9) is a classic one and has been used in prior works (Tsakiris & Vidal, 2018a; Ding et al., 2019b; 2020a; Iwata et al., 2020; Kümmerle et al., 2021; Peng et al., 2022b). While we have not seen the appearance of (10) in its exact form in the literature, several papers have considered (10) with particualr $\mathcal{M}$'s for the problems of robust subspace recovery (Tsakiris & Vidal, 2018a; Ding et al., 2019b; 2020a) and essential matrix estimation (Zhao et al., 2020; Zhao, 2022). In the former case, (10) is a weighted eigenvalue problem, easily solvable; in the latter case, (10) is a QCQP, which can be solved via its SDP relaxations. Since (10) does not involve binary decision variables (different from Lajoie et al. (2018); Yang & Carlone (2019; 2022); Peng et al. (2022a)), the SDP solver of Zhao et al. (2020); Zhao (2022) is actually efficient. Given these, we implicitly assume that (10) is efficiently solvable, and later we will analyze the convergence rate of IRLS in generality and evaluate its performance in action.

## 4 THEORETICAL ANALYSIS

In this section, we first characterize the geometry of inliers, outliers, and their interplay with $\mathcal{M}$ via some quantities (§4.1). They turn out to be useful in the analysis of the minimizers (§4.2) of and algorithms (§4.3) for ($OR^2$-$\ell^1$).

### 4.1 GEOMETRIC QUANTITIES

We will define quantities that depend on the inliers $\mathcal{X}$, outliers $\mathcal{O}$, the true vector $\boldsymbol{b}^*$ (equivalently, the true hyperplane $\mathcal{H}$), and the manifold $\mathcal{M}$. It should be clear that these quantities are for theoretical understanding and not for running the algorithms. To begin with, define

$$c_{\mathrm{in}} := \inf\left\{ \frac{1}{N} \sum_{j=1}^{N} |\boldsymbol{x}_j^\top \boldsymbol{s}| : \boldsymbol{b} \in \mathcal{M},\ \boldsymbol{b} \neq \boldsymbol{b}^*,\ \boldsymbol{s} = \boldsymbol{\Pi}_{\mathbb{S}^{D-1}}(\boldsymbol{\Pi}_{\mathcal{H}}(\boldsymbol{b})) \right\}. \qquad (11)$$

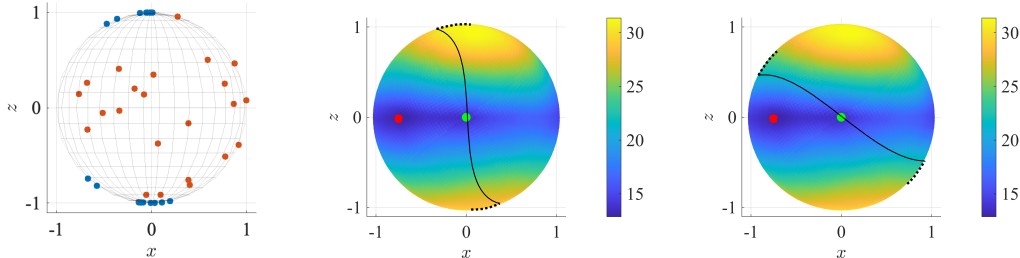

Figure 2: The non-uniform data from Figure 1 (left), and $f(\cdot)$ over $\mathbb{S}^2$ using the data overlaid by two $\mathcal{M}$'s (dark solid lines) yielding large $c_{\mathrm{in}}$ (middle) and small $c_{\mathrm{in}}$ (right) respectively. Dotted lines are locations of $\boldsymbol{s}$'s in $c_{\mathrm{in}}$ (11). For clarity, the viewpoints are set at the true $\boldsymbol{b}^*$ facing towards the origin.

*Remark* 3. To understand $c_{\mathrm{in}}$, note that one can expect it to be large when for any $\boldsymbol{b} \in \mathcal{M}, \boldsymbol{b} \neq \boldsymbol{b}^*$, its projection $\boldsymbol{s}$ onto $\mathcal{H}$ is *not* orthogonal to most inliers $\boldsymbol{x}_j$'s. This is visualized in Figure 2, where $c_{\mathrm{in}}$ is large as the dotted lines are far from the equator $z = 0$ (middle), and vice versa (right). To better appreciate the role of $\mathcal{M}$ in $c_{\mathrm{in}}$, we make a contrast with a similar quantity in Zhu et al. (2018) for guaranteeing (1):

$$\bar{c}_{\mathrm{in}} = \inf\left\{\frac{1}{N}\sum_{j=1}^{N}|\boldsymbol{x}_j^\top \boldsymbol{s}| : \boldsymbol{s} \in \mathcal{H} \cap \mathbb{S}^{D-1}\right\}. \tag{12}$$

Observe that $\bar{c}_{\mathrm{in}}$ is large only when *all directions* in $\mathcal{H}$ are not orthogonal to most inliers, essentially requiring inliers to be uniformly distributed in $\mathcal{H} \cap \mathbb{S}^{D-1}$. Such an assumption can fail in applications as illustrated in Example 1. In contrast, one can have large $c_{\mathrm{in}}$ albeit some vectors in $\mathcal{H}$ are orthogonal to a vast number of inliers, as long as these directions are ruled out by $\mathcal{M}$.

*Example* 1. In geometric vision applications, inliers tend to lie close to a *proper* subspace of the true hyperplane. For instance, in *essential matrix* or *trifocal tensor estimation*, data resulting from 3D points on an affine plane provably lies in a subspace of lower dimension (Chum et al., 2005; Ding et al., 2020b); this is more generally known as critical surfaces (Hartley & Zisserman, 2003, §22).

With the above said, we can further define similar quantities for outliers:

$$c_{\mathrm{out,min}} := \min_{\boldsymbol{b} \in \mathbb{S}^{D-1}} \frac{1}{M}\sum_{j=1}^{M}|\boldsymbol{o}_j^\top \boldsymbol{b}|, \quad c_{\mathrm{out,max}} := \max_{\boldsymbol{b} \in \mathbb{S}^{D-1}} \frac{1}{M}\sum_{j=1}^{M}|\boldsymbol{o}_j^\top \boldsymbol{b}|. \tag{13}$$

*Remark* 4. It is now not hard to see that $c_{\mathrm{out,min}}$ and $c_{\mathrm{out,max}}$ are close if the outliers are uniformly distributed in $\mathbb{S}^{D-1}$. While in principle one can consider the $\mathcal{M}$-constrained version of these outlier-related quantities, we find doing so complicates the picture without shedding useful insights.

Finally, we define

$$\gamma_{\mathrm{in},\theta} := \max\left\{\frac{1}{N}\left\|\boldsymbol{\Pi}_{T_{\boldsymbol{b}}\mathcal{M}^\perp \cap \langle b \rangle^\perp}\sum_{j=1}^{N}\mathrm{sgn}(\boldsymbol{x}_j^\top \boldsymbol{b})\boldsymbol{x}_j\right\|_2 : \boldsymbol{b} \in \mathcal{B}_{\mathcal{M}}(\boldsymbol{b}^*,\theta)\right\}, \tag{14}$$

$$\bar{\eta}_{\mathrm{out}} := \max_{\boldsymbol{b} \in \mathbb{S}^{D-1}} \frac{1}{M}\left(\left\|\boldsymbol{\Pi}_{T_{\boldsymbol{b}}\mathcal{M}}\sum_{j=1}^{M}\mathrm{sign}(\boldsymbol{o}_j^\top \boldsymbol{b})\boldsymbol{o}_j\right\|_2 + D\right), \tag{15}$$

where $\mathcal{B}_{\mathcal{M}}(\boldsymbol{b}^*,\theta)$ is a ball of angular radius $\theta$ centered at $\boldsymbol{b}^*$ over $\mathcal{M}$

$$\mathcal{B}_{\mathcal{M}}(\boldsymbol{b}^*,\theta) := \{\boldsymbol{b} \in \mathcal{M} : \angle(\boldsymbol{b},\boldsymbol{b}^*) \leq \theta\}. \tag{16}$$

*Remark* 5. Roughly speaking, $T_{\boldsymbol{b}}\mathcal{M}^\perp \cap \langle b \rangle^\perp$ contains directions *tangent* to $\mathbb{S}^{D-1}$ and *normal* to $\mathcal{M}$ at $\boldsymbol{b}$, as seen by the arrows in Figure 3. As such, $\gamma_{\mathrm{in},\theta}$ is small if within $\mathcal{B}_{\mathcal{M}}(\boldsymbol{b}^*,\theta)$, all such directions are orthogonal to most inliers $\boldsymbol{x}_j$'s; this is, e.g., the case for the left and middle plots in Figure 3 with $\theta = \frac{\pi}{2}$ and $\theta = \frac{\pi}{3}$ respectively, but *not* the case for the right plot with $\theta = \frac{\pi}{3}$. Observe also that $\gamma_{\mathrm{in},\theta}$ is increasing in $\theta$. Last but not least, $\bar{\eta}_{\mathrm{out}}$ is small if outliers are uniformly distributed in $\mathbb{S}^{D-1}$.

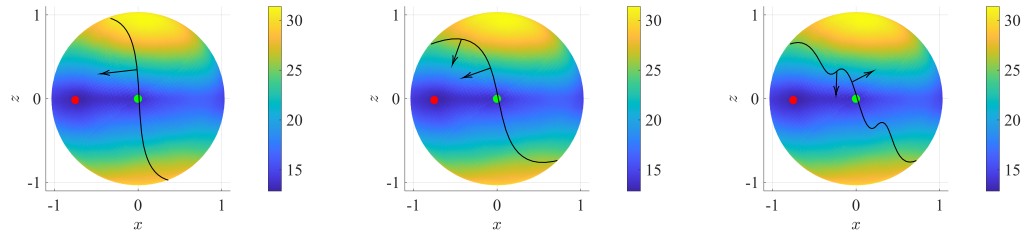

Figure 3: $f(\cdot)$ over $\mathbb{S}^2$ using the non-uniform data from Figure 1 overlaid by three $\mathcal{M}$'s (dark solid lines) yielding $\gamma_{\text{in},\theta}$ globally small (left), locally small (middle) and not small (right) respectively. Arrows are examples of vectors from $T_{\boldsymbol{b}}\mathcal{M}^\perp \cap \langle\boldsymbol{b}\rangle^\perp$ in $\gamma_{\text{in},\theta}$ (14).

### 4.2 GUARANTEES ON GLOBAL MINIMIZERS

Using the quantities developed above, we are now ready to localize global minimizers of (OR$^2$-$\ell^1$). To begin with, when $\gamma_{\text{in},\theta}$ is small for all $\theta \in (0, \frac{\pi}{2}]$ (e.g., Figure 3 left), we have the theorem below:

**Theorem 1.** *Let $\gamma_{\text{in}} := \sup_{\theta\in(0,\frac{\pi}{2}]} \gamma_{\text{in},\theta}$. Every global minimizer of (OR$^2$-$\ell^1$) must be $\pm\boldsymbol{b}^*$ if*

$$\frac{\frac{M^2}{N^2} \cdot \left[\left(c_{\text{out,max}} - c_{\text{out,min}}\right)^2 + \overline{\eta}_{\text{out}}^2\right] + \gamma_{\text{in}}^2}{c_{\text{in}}^2} < 1. \tag{17}$$

Intuitively, Theorem 1 says the minimizers of (OR$^2$-$\ell^1$) recover the ground truth $\pm\boldsymbol{b}^*$ as long as $N \gg M$, inliers are well distributed *with respect to* $\mathcal{M}$ ($c_{\text{in}}$ large), outliers are well distributed ($c_{\text{out,max}} - c_{\text{out,min}}$ and $\overline{\eta}_{\text{out}}$ small), and $\mathcal{M}$ does not curve too much against the inliers ($\gamma_{\text{in}}$ small).

On the other hand, $\gamma_{\text{in},\theta}$ could be small only for small $\theta$ and increases drastically for large $\theta$ (e.g., Figure 3 middle). In this case, we provide the following guarantee:

**Theorem 2.** *Every global minimizer of (OR$^2$-$\ell^1$) must be $\pm\boldsymbol{b}^*$ if*

$$\sin(\theta) > \frac{M}{N} \cdot \frac{c_{\text{out,max}} - c_{\text{out,min}}}{c_{\text{in}}}, \tag{18}$$

$$N^2 c_{\text{in}}^2 - N^2 \gamma_{\text{in},0}^2 \geq M^2 \overline{\eta}_{\text{out}}^2, \tag{19}$$

*where $\theta$ is the unique solution to*

$$\cos^2(\theta) N^2 c_{\text{in}}^2 - N^2 \gamma_{\text{in},\theta}^2 = M^2 \overline{\eta}_{\text{out}}^2. \tag{20}$$

Note that the LHS of (20) is strictly decreasing in $\theta$, and non-positive with $\theta = \frac{\pi}{2}$; the RHS of (20) is non-negative. Thus, with condition (19) in place, (20) is guaranteed to have a unique solution. Theorem 2 is tighter than Theorem 1: with some inspection one can see that the former entails the latter; if for some $\theta = \overline{\theta}$ (18) holds and in (20) the LHS is greater than or equal to the RHS, then the minimizers of (OR$^2$-$\ell^1$) are $\pm\boldsymbol{b}^*$ *regardless of* the behavior of $\gamma_{\text{in},\theta}$ for $\theta > \overline{\theta}$.

Finally, if $\gamma_{\text{in},\theta}$ is large even for small $\theta$ (e.g., Figure 3 right), we can only characterize the approximate location of the minimizer with the following theorem that does not depend on $\gamma_{\text{in},\theta}$.

**Theorem 3.** *Every global minimizer of (OR$^2$-$\ell^1$) has an angle away from $\pm\boldsymbol{b}^*$ of at most*

$$\arcsin\left(\frac{M}{N} \cdot \frac{c_{\text{out,max}} - c_{\text{out,min}}}{c_{\text{in}}}\right). \tag{21}$$

### 4.3 CONVERGENCE RATES OF RSGM AND IRLS

We now consider convergence analysis of the proposed algorithms. We begin with Algorithm 1:

**Theorem 4.** *With a suitable initialization and geometrically diminishing step size, the sequence $\{\boldsymbol{b}_k\}_{k=0}^K$ given by Algorithm 1 converges at a linear rate to the true vector $\pm\boldsymbol{b}^*$. More precisely, with the step size $\alpha_k = \beta^k \cdot \alpha_0$, the principal angle $\theta_k$ between $\boldsymbol{b}_k$ and $\boldsymbol{b}^*$ is upper bounded as per*

$$\sin(\theta_k/2) \leq \beta^k \sin(\theta_0/2) \tag{22}$$

*if the following conditions are satisfied:*

*(i) Initialization not too far from $\pm\boldsymbol{b}^*$: $\sin(\theta_0/2) < \frac{s}{2c_2}$,*

*(ii) Small initial step size: $0 < \alpha_0 < \min\left(\frac{\sin(\theta_0/2)}{s - 2c_2\sin(\theta_0/2)}, \frac{4(s - 2c_2\sin(\theta_0/2))}{(c_1+1)\kappa^2/\sin(\theta_0/2)}\right)$,*

*(iii) Suitable step size decay: $1 > \beta \geq \sqrt{1 + \alpha_0\left(\alpha_0\frac{(c_1+1)\kappa^2}{4\sin(\theta_0/2)^2} + 2c_2 - \frac{s}{\sin(\theta_0/2)}\right)}$.*

*In the above, $s := \frac{1}{2}Nc_{\mathrm{in}} - Mc_{\mathrm{out,max}}$, $\kappa$ is a Lipschitz constant of the objective $f(\cdot)$, $c_2 = \kappa \cdot c_3$, and $c_1$ and $c_3$ are constants that depend only on $\mathcal{M}$.*

Assuming condition (i) holds, one can verify that conditions (ii) and (iii) of Theorem 4 also holds true if the initial step size $\alpha_0$ is small enough as $\beta \to 1$. Therefore, with $\alpha_0$ and $\beta$ suitably chosen, Algorithm 1 converges linearly if initialized in a neighborhood of $\boldsymbol{b}^*$ of size $s/c_2$ (Theorem 4).

We then proceed to Algorithm 2, an IRLS method of an entirely different style than Algorithm 1. The theoretical convergence rate guarantee, stated below, is also in a different style than Theorem 4:

**Theorem 5.** *The sequence $\{\boldsymbol{b}_k\}_{k=0}^K$ produced by Algorithm 2 converges at a sublinear rate to a critical point of the following problem:*

$$\min_{\boldsymbol{b} \in \mathcal{M}} \sum_{j=1}^{L} h(\boldsymbol{y}_j^\top \boldsymbol{b}), \qquad h(r) := \begin{cases} |r| & |r| > \delta \\ \frac{1}{2}\left(r^2/\delta + \delta\right) & |r| \leq \delta \end{cases} \qquad \text{(OR}^2\text{-Huber)}$$

*More precisely, the Riemannian gradient of the objective of (OR$^2$-Huber) is bounded above:*

$$\min_{k=0,\dots,K} \left\| \mathrm{rgrad} \sum_{j=1}^{L} h(\boldsymbol{y}_j^\top \boldsymbol{b}_k) \right\|_2 \leq \sqrt{\frac{2\sigma \cdot \sum_{j=1}^{L} h(\boldsymbol{y}_j^\top \boldsymbol{b}_0)}{K+1}} \qquad (23)$$

*In (23), $\sigma$ is a finite positive constant depending on the manifold $\mathcal{M}$, data $\boldsymbol{y}_j$'s, and $\delta$.*

Algorithm 2 converges to a critical point of a different Huber-tyle objective in (OR$^2$-Huber) than the $\ell^1$ objective of (OR$^2$-$\ell^1$). The key is that the (rescaled and translated) Huber loss $h(\cdot)$ in (OR$^2$-Huber) tends to the $\ell^1$ loss as $\delta \to 0$, and (OR$^2$-Huber) can be thought of as a smoothed version of (OR$^2$-$\ell^1$), easier to minimize. Theorem 5 reveals Algorithm 2 performs such smoothing implicitly with a sublinear convergence rate. Our proof of Theorem 5 extends the idea of Hong et al. (2017), which is specifically for the convex case, into the case with non-convex manifolds $\mathcal{M}$.

## 5 EXPERIMENTS: ROBUST ESSENTIAL MATRIX ESTIMATION

Recall that our major claim is that enforcing manifold constraints would lead to theoretically more meaningful and practically more robust estimation than relaxing, if not ignoring, the manifold constraints. While previous sections focus on providing theoretical insights, here we are interested in consolidating our claim using numerical experiments. In particular, this section presents experimental results on robust essential matrix estimation, while in the interest of space we refer the reader to the Appendix for more experiments and details.

In order to validate our claim, we consider the following setup. We follow Kneip & Furgale (2014) to generate random rotations $\boldsymbol{R}^* \in \mathbb{SO}(3)$, translations $\boldsymbol{t}^* \in \mathbb{R}^3$, and $L$ correspondences, of which $N$ are inliers and $M$ are outliers. The correspondences are used to produce $\mathcal{Y} \in \mathbb{R}^{9 \times L}$ following (4) and a normalization of columns. We test the IRLS algorithm for solving (1) and (OR$^2$-$\ell^1$). For (1), the weighted least-squares step (10) can be implemented via SVD; the resulting IRLS algorithm was proposed in Tsakiris & Vidal (2018b) and here we call it DPCP[3] (*dual principal component pursuit*). For (OR$^2$-$\ell^1$), with $\mathcal{M}$ being the essential manifold $\mathcal{M}_E$, the weighted least-squares step (10) can be implemented via an efficient SDP solver (Zhao, 2022); here we call it OR$^2$. Each method is run on $\mathcal{Y}$ to obtain an estimate $\hat{\boldsymbol{b}}$, while we need to project the estimate of DPCP onto $\mathcal{M}_E$ as it does not enforce the essential manifold constraint. We compute the angular error between $\hat{\boldsymbol{b}}$ and the ground truth $\boldsymbol{b}^*$. We further decompose each estimated essential matrix into rotations and translations, and calculate angular rotation and translation error with $\boldsymbol{R}^*, \boldsymbol{t}^*$.

---

[3]We refer the reader to Tsakiris & Vidal (2018b); Ding et al. (2020b); Zhao (2022), where DPCP or IRLS really stood out with excellent performance in comparison to other competitive methods including RANSAC.

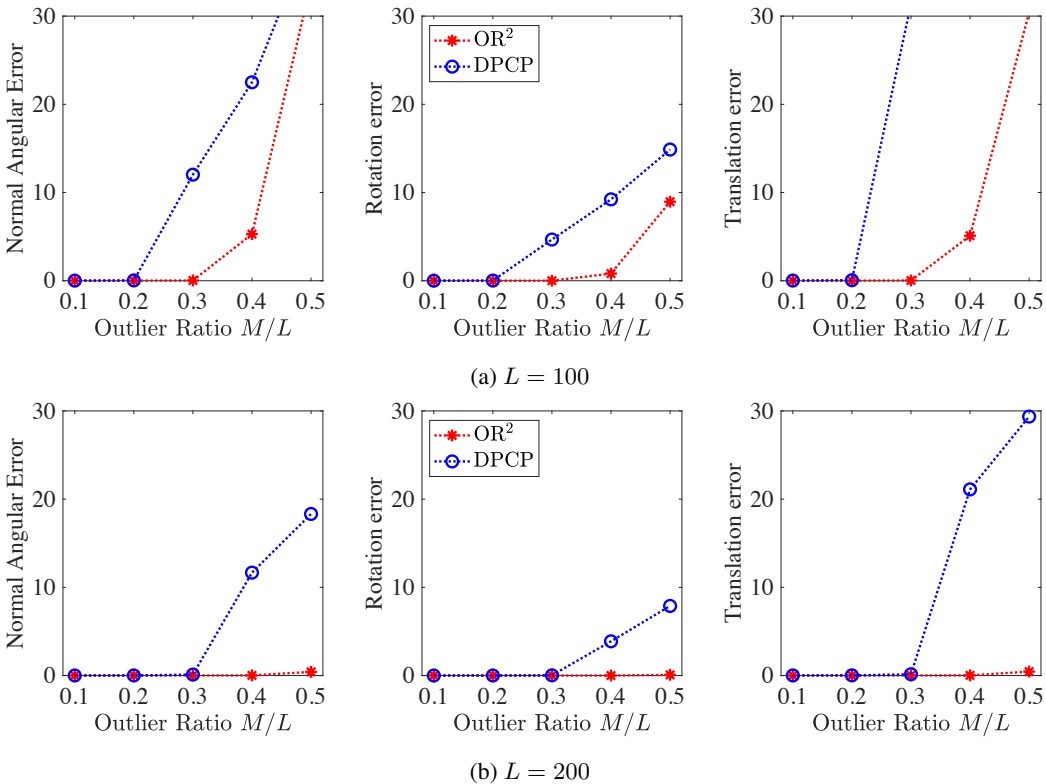

Figure 4: Robust essential matrix estimation over 50 randomly generated scene data.

In Figure 4 we report the median errors for varying outlier ratio $M/L \in \{0.1 : 0.1 : 0.5\}$ and total number of points $L \in \{100, 200\}$. As expected, the two methods, DPCP and OR$^2$, give almost 0 error with a small (e.g., $\leq 20\%$) outlier ratio. Notably, also can be seen is that (OR$^2$-$\ell^1$) is much more robust than (1): when $L = 100$, (OR$^2$-$\ell^1$) has a translation error of $5.2°$ with $40\%$ outliers, while (1) yields that of more than $30°$ with only $30\%$ outliers, a sharp contrast. Moreover, with more points ($L = 200$), (OR$^2$-$\ell^1$) achieves less than $1°$ for any error metric even with $50\%$ outliers, while in that case (1) has, e.g., $7.8°$ rotation error already. The experiments therefore confirm our claim that directly tackling the manifold constraints (e.g., essential manifolds) yield visible computational benefits than relaxing the constraints (e.g., into a unit sphere).

## 6 CONCLUSION AND DISCUSSION

In this paper, we presented a meaningful generalization of the seminal work of Späth & Watson (1987) on $\ell^1$ orthogonal regression (on the sphere) to outlier-robust orthogonal regression on smooth manifolds, thereby bringing together robustness with the proper manifold constraints that are inherent in geometric vision (e.g., essential manifolds). Our experiments suggested that taking manifold constraints into consideration does improve the performance of the estimation algorithms. Even though handling manifold constraints directly is arguably more challenging, our core contribution consists of performing theoretical analysis for the proposed approach and derived several results regarding the optimization landscape and convergence of associated algorithms; all these are ultimately useful guarantees for outlier-robust orthogonal regression on manifolds.

Our analysis is not without limitations. First of all, to obtain conditions for global minimizers (§4.2), we defined several quantities in §4.1 that are not directly computable. However, for future work it is possible to derive probabilistic bounds for them, thereby obtaining more informative conditions. The second, related, question to consider is on specializing our conditions for manifolds of interest (e.g., essential manifolds). Finally, we focused on orthogonal regression in the paper which is associated with the $\ell^1$ (and Huber) loss. It is an important future direction to generalize our analysis for other common loss functions (e.g., Logcosh, Geman-McClure, truncated quadratic).

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
