# A  MORE ON EXPERIMENTS

## A.1  DATA GENERATION FOR ESSENTIAL MATRIX ESTIMATION (§5)

We first generate the ground truth rotation $\boldsymbol{R}^*$ and translation $\boldsymbol{t}^*$. $\boldsymbol{t}^*$ is sampled uniformly at random from $[-2, 2]^3$. $\boldsymbol{R}^*$ is generated corresponding to rotating about the $x$-axis for $\theta_x$ radians, the $y$-axis for $\theta_y$ radians, and the $z$-axis for $\theta_z$ radians; $\theta_x, \theta_y, \theta_z$ are sampled independently and uniformly at random from $[-0.5, 0.5]$. A total of $L$ points in $\mathbb{R}^3$ are randomly sampled using the following procedure:

---
**Algorithm 3:** 3D Point Cloud Generation

---
1 Number of points: $L$
2 $d_{\min} \leftarrow 4, d_{\max} \leftarrow 8$
3 For $j \leftarrow 1, \ldots, L$:

$$\boldsymbol{p}_j \sim \mathrm{unif}([-1, 1]^3) \tag{24}$$
$$\boldsymbol{d}_j \leftarrow \boldsymbol{\Pi}_{\mathbb{R}^2}(\boldsymbol{p}_j) \tag{25}$$
$$\boldsymbol{q}_j \leftarrow (d_{\max} - d_{\min})\boldsymbol{p}_j + d_{\min}\boldsymbol{d}_j \tag{26}$$

4 **return** Generated 3D points $\boldsymbol{Q}$

---

Roughly speaking, each point is randomly sampled to have a distance $4-8$ around the origin. These points in $\mathbb{R}^3$ are projected onto the cameras defined by $[\boldsymbol{I}, \boldsymbol{0}]$ and $[\boldsymbol{R}^*, \boldsymbol{t}^*]$ to give $L$ correspondences. We take $M$ of them and randomly permute the correspondence relations to produce outliers. Then, as said in §5, the correspondences are used to produce $\mathcal{Y} \in \mathbb{R}^{9 \times L}$ following (4) and a projection onto $\mathbb{S}^8$ for normalization.

## A.2  ROBUST ROTATION ESTIMATION

**Problem.** Given a point cloud in $\mathbb{R}^3$ and its copy that undergoes a rotation $\boldsymbol{R}^* \in \mathbb{SO}(3)$, how can we recover $\boldsymbol{R}^*$? Again, one can first run some matching algorithm on the point clouds to obtain correspondences $\{(\boldsymbol{z}_j, \boldsymbol{z}_j')\}_{j=1}^L \subset \mathbb{R}^3 \times \mathbb{R}^3$. Inlier correspondences $(\boldsymbol{z}_j, \boldsymbol{z}_j')$ now satisfy

$$\boldsymbol{z}_j' = \boldsymbol{R}^* \boldsymbol{z}_j, \tag{27}$$

while outlier correspondences exist due to false matches. Our focus is to robustly estimate $\boldsymbol{R}^*$ given $\{(\boldsymbol{z}_j, \boldsymbol{z}_j')\}_{j=1}^L$. Being able to estimate $\boldsymbol{R}^*$ accurately is crucial to a more challenging task of *point cloud registration*, where inlier correspondences $(\boldsymbol{z}_j, \boldsymbol{z}_j')$ further differ by a translation $\boldsymbol{t}^*$, that is $\boldsymbol{z}_j' = \boldsymbol{R}^* \boldsymbol{z}_j + \boldsymbol{t}^*$.

**OR$^2$ over Rotation Matrices.** To see that this problem can be posed as an instance of OR$^2$, observe that (27) implies

$$[\boldsymbol{z}_j']_\times \boldsymbol{R}^* \boldsymbol{z}_j = \boldsymbol{0} \in \mathbb{R}^3, \tag{28}$$

by noting $\boldsymbol{a} \times \boldsymbol{a} = \boldsymbol{0}$ for any $\boldsymbol{a} \in \mathbb{R}^3$ and [1]. These are three equations linear in $\boldsymbol{R}^*$, only two of which are linearly independent. Thus one can write (28) as

$$\boldsymbol{Y}_j^\top \boldsymbol{b}^* = \boldsymbol{0} \in \mathbb{R}^2, \tag{29}$$

where $\boldsymbol{Y}_j \in \mathbb{R}^{9 \times 2}$ is some embedding of $(\boldsymbol{z}_j, \boldsymbol{z}_j')$, and $\boldsymbol{b}^* = \mathrm{vec}(\boldsymbol{R}^*)/\sqrt{3}$ lies in the manifold

$$\mathcal{M}_R := \{\boldsymbol{b} \in \mathbb{R}^9 : \quad \boldsymbol{b} = \mathrm{vec}(\boldsymbol{R})/\sqrt{3}, \quad \boldsymbol{R} \in \mathbb{SO}(3)\}, \tag{30}$$

Since $\boldsymbol{Y}_j$ is an inlier when *both* of its columns are orthogonal to $\boldsymbol{b}^*$, we consider a natural extention[4] of (OR$^2$-$\ell^1$)

$$\min_{\boldsymbol{b}} \quad \sum_{j=1}^L \left\| \boldsymbol{Y}_j^\top \boldsymbol{b} \right\|_2 \quad \text{s.t.} \quad \boldsymbol{b} \in \mathcal{M} = \mathcal{M}_R. \tag{OR$^2$-$\ell^1$-group}$$

---
[4]Indeed, it is easy to verify that (OR$^2$-$\ell^1$-group) reduces to (OR$^2$-$\ell^1$) if each $\boldsymbol{Y}_j$ only has one column.

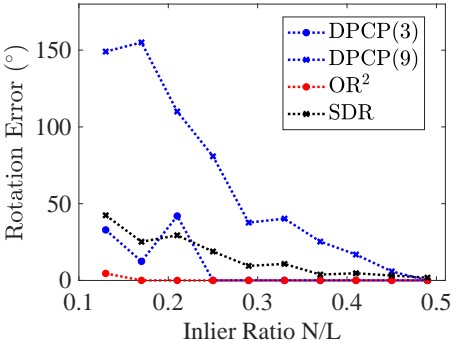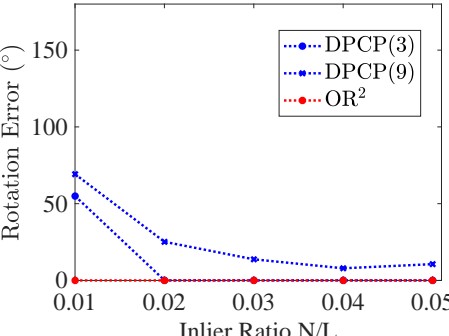

Figure 5: Robust rotation search with varying number of inliers and total number of points $L = 100$ (left) and $L = 10000$ (right). The SDR method fail to run in the latter case due to insufficient memory.

To normalize the data, each column of $\boldsymbol{Y}_j$ is scaled to have a unit $\ell^2$ norm. We consider an RSGM algorithm for (OR$^2$-$\ell^1$-group), in which the projection map onto $T_{\boldsymbol{b}}\mathcal{M}$ and the retraction map are implemented by the manopt toolbox (Boumal et al., 2013).

**Relaxation to $\mathbb{S}^8$.** Similar to the case of robustly estimating an essential matrix (§1), since $\mathcal{M}_R \subset \mathbb{S}^8$ one can relax (OR$^2$-$\ell^1$-group) to

$$\min_{\boldsymbol{b}} \quad \sum_{j=1}^{L} \left\| \boldsymbol{Y}_j^\top \boldsymbol{b} \right\|_2 \quad \text{s.t.} \quad \boldsymbol{b} \in \mathcal{M} = \mathbb{S}^8. \tag{31}$$

This special instance has been considered in the related problem of homography estimation (Ding et al., 2020b). We use an RSGM algorithm for solving (31), and refer to such a method as DPCP(9) (Zhu et al., 2018). The number 9 represents the coordinate dimension of $\boldsymbol{b}^*$. As one might expect, we also have a baseline that operates in a different dimension.

**Relaxation to $\mathbb{S}^2$.** Let $\overline{\boldsymbol{b}} \in \mathbb{S}^2$ be the rotation axis of $\boldsymbol{R}^*$. Applying dot product on (27) with $\overline{\boldsymbol{b}}$ yields

$$\langle \overline{\boldsymbol{b}}, \boldsymbol{z}_j' \rangle = \langle \overline{\boldsymbol{b}}, \boldsymbol{R}^* \boldsymbol{z}_j \rangle = \langle \overline{\boldsymbol{b}}, \boldsymbol{z}_j \rangle \tag{32}$$

$$\Leftrightarrow \langle \overline{\boldsymbol{b}}, \boldsymbol{z}_j' - \boldsymbol{z}_j \rangle = 0. \tag{33}$$

Therefore, we can estimate $\overline{\boldsymbol{b}}$ by solving (1) with $\boldsymbol{y}_j$ taken as $(\boldsymbol{z}_j' - \boldsymbol{z}_j)/\left\| \boldsymbol{z}_j' - \boldsymbol{z}_j \right\|_2$ and $D = 3$. Once we have a solution, say $\hat{\boldsymbol{b}}$, we identify a set of inliers by thresholding the error $|\langle \hat{\boldsymbol{b}}, \boldsymbol{y}_j \rangle|$ by $10^{-3}$. Given the estimated rotation axis $\hat{\boldsymbol{b}}$ and estimated inliers, one can compute the rotation angle via interval stabbing as in Peng et al. (2022c), after which a rotation estimation can be formed by composing the estimated axis and angle. We call such an approach DPCP(3) in the experiments.

**Semidefinite Programming.** Finally, we compare with a truncated least squares surrogate of (OR$^2$-$\ell^0$) with $\mathcal{M} = \mathcal{M}_R$, which is solved via a *Shor* semidefinite relaxation (Yang & Carlone, 2019; Peng et al., 2022a); see also §2. In our experiments this is referred to as SDR.

**Data Generation.** To generate the ground-truth rotation $\boldsymbol{R}^*$, we sample uniformly at random an axis from $\mathbb{S}^2$ and an angle from $[0, 2\pi]$, and form $\boldsymbol{R}^*$ by composing the axis and angle. We sample a point cloud of $N$ points from $\mathcal{N}(\boldsymbol{0}, \boldsymbol{I}_3)$, and rotate them by $\boldsymbol{R}^*$ to give the inlier correspondences. To generate outlier correspondences, we sample *two* point clouds of $M$ points from $\mathcal{N}(\boldsymbol{0}, \boldsymbol{I}_3)$, and scale the points from the second point cloud such that $\|\boldsymbol{z}_j'\|_2 = \|\boldsymbol{z}_j\|_2$; otherwise the outliers can be easily removed by comparing the norms.

**Result.** Figure 5 reports the median rotation error in degrees over 10 random seeds with varying number of points and outlier ratio. Firstly, in all cases OR$^2$ has a lower error than DPCP (the red curve stays below all other curves), as also observed in estimating essential matrix §5. Notably,

when $L = 100$, $\text{OR}^2$ is more accurate than SDR, the latter being a state-of-the-art approach (Yang & Carlone, 2019). Moving on to the regime of large $L = 10,000$, the SDR method fails due to the prohibitive memory required to handle $O(L^2)$ variables. In contrast, $\text{OR}^2$ continue to give the most accurate estimate: even with $1\%$ inliers it gives close to $0°$ median error.

## B  PROOFS

### B.1  PROOFS OF MINIMIZERS

We begin this section by giving a lemma on the critical points which stands as a building block in proving Theorems 1 and 2.

**Lemma 6.** *Any critical point of* $(\text{OR}^2\text{-}\ell^1)$ *is either* $\pm b^*$ *or has an angle of at least* $\theta$ *away from* $\pm b^*$, *where* $\theta$ *is the solution to*

$$\cos^2(\theta)N^2 c_{\text{in}}^2 - N^2\gamma_{\text{in},\theta}^2 = M^2\bar{\eta}_{\text{out}}^2. \tag{34}$$

*Proof.* Let $b$ be a critical point of $(\text{OR}^2\text{-}\ell^1)$. We are done if it is $\pm b^*$, so suppose the otherwise. Note that $b$ by definition satisfies the following subgradient condition

$$\mathbf{0} = (\boldsymbol{I} - \boldsymbol{b}\boldsymbol{b}^\top - \boldsymbol{Q}\boldsymbol{Q}^\top)\left(\sum_{j=1}^{N} \text{sgn}(\boldsymbol{x}_j^\top\boldsymbol{b})\boldsymbol{x}_j + \sum_{j=1}^{M} \text{sgn}(\boldsymbol{o}_j^\top\boldsymbol{b})\boldsymbol{o}_j\right), \tag{35}$$

where $\boldsymbol{I} - \boldsymbol{b}\boldsymbol{b}^\top - \boldsymbol{Q}\boldsymbol{Q}^\top$ is the orthogonal projection onto $T_{\boldsymbol{b}}\mathcal{M}$ such that $\boldsymbol{Q}^\top\boldsymbol{Q} = \boldsymbol{I}$ and $\boldsymbol{Q}^\top\boldsymbol{b} = \boldsymbol{0}$. We will now bound the 'size' of the subgradients for inliers and outliers respectively.

**Lower bounding the size of subgradient for inliers.** Define $\theta$ to be the angle between $b$ and $b^*$. Since $b \subset \mathcal{M} \subset \mathbb{S}^{D-1}$ and $b \not\perp \mathcal{H}$, one write $b$ as

$$\boldsymbol{b} = \sin(\theta)\boldsymbol{s} + \cos(\theta)\boldsymbol{n}, \tag{36}$$

where $\boldsymbol{s} := \boldsymbol{\Pi}_{\mathcal{H}}(\boldsymbol{b})/\|\boldsymbol{\Pi}_{\mathcal{H}}(\boldsymbol{b})\|_2$, $\boldsymbol{n} := \boldsymbol{\Pi}_{\mathcal{H}^\perp}(\boldsymbol{b})/\|\boldsymbol{\Pi}_{\mathcal{H}^\perp}(\boldsymbol{b})\|_2$, and . Now,

$$\left\| (\boldsymbol{I} - \boldsymbol{b}\boldsymbol{b}^\top - \boldsymbol{Q}\boldsymbol{Q}^\top) \sum_{j=1}^{N} \mathrm{sgn}(\boldsymbol{x}_j^\top \boldsymbol{b})\boldsymbol{x}_j \right\|_2^2 \tag{37}$$

$$\stackrel{(36)}{=} \left\| (\boldsymbol{I} - \boldsymbol{Q}\boldsymbol{Q}^\top) \sum_{j=1}^{N} \mathrm{sgn}(\boldsymbol{x}_j^\top \boldsymbol{b})\boldsymbol{x}_j - \sin(\theta)\boldsymbol{b} \sum_{j=1}^{N} \mathrm{sgn}(\boldsymbol{x}_j^\top \boldsymbol{s})(\boldsymbol{x}_j^\top \boldsymbol{s}) \right\|_2^2 \tag{38}$$

$$\stackrel{\|\boldsymbol{b}\|=1}{=} \left\| (\boldsymbol{I} - \boldsymbol{Q}\boldsymbol{Q}^\top) \sum_{j=1}^{N} \mathrm{sgn}(\boldsymbol{x}_j^\top \boldsymbol{b})\boldsymbol{x}_j \right\|_2^2 + \sin^2(\theta) \left\| \sum_{j=1}^{N} \mathrm{sgn}(\boldsymbol{x}_j^\top \boldsymbol{s})(\boldsymbol{x}_j^\top \boldsymbol{s}) \right\|_2^2$$
$$- 2 \left\langle (\boldsymbol{I} - \boldsymbol{Q}\boldsymbol{Q}^\top) \sum_{j=1}^{N} \mathrm{sgn}(\boldsymbol{x}_j^\top \boldsymbol{b})\boldsymbol{x}_j, \boldsymbol{b} \sum_{j=1}^{N} \mathrm{sgn}(\boldsymbol{x}_j^\top \boldsymbol{s})(\boldsymbol{x}_j^\top \boldsymbol{s}) \right\rangle \tag{39}$$

$$\stackrel{\boldsymbol{Q}^\top \boldsymbol{b}=\boldsymbol{0}}{=} \left\| (\boldsymbol{I} - \boldsymbol{Q}\boldsymbol{Q}^\top) \sum_{j=1}^{N} \mathrm{sgn}(\boldsymbol{x}_j^\top \boldsymbol{b})\boldsymbol{x}_j \right\|_2^2 + \sin^2(\theta) \left\| \sum_{j=1}^{N} \mathrm{sgn}(\boldsymbol{x}_j^\top \boldsymbol{s})(\boldsymbol{x}_j^\top \boldsymbol{s}) \right\|_2^2$$
$$- 2 \left\langle \sum_{j=1}^{N} \mathrm{sgn}(\boldsymbol{x}_j^\top \boldsymbol{b})\boldsymbol{x}_j, \boldsymbol{b} \sum_{j=1}^{N} \mathrm{sgn}(\boldsymbol{x}_j^\top \boldsymbol{s})(\boldsymbol{x}_j^\top \boldsymbol{s}) \right\rangle \tag{40}$$

$$\stackrel{(36)}{=} \left\| (\boldsymbol{I} - \boldsymbol{Q}\boldsymbol{Q}^\top) \sum_{j=1}^{N} \mathrm{sgn}(\boldsymbol{x}_j^\top \boldsymbol{b})\boldsymbol{x}_j \right\|_2^2 + \sin^2(\theta) \left\| \sum_{j=1}^{N} \mathrm{sgn}(\boldsymbol{x}_j^\top \boldsymbol{s})(\boldsymbol{x}_j^\top \boldsymbol{s}) \right\|_2^2$$
$$- 2\sin^2(\theta) \left\| \sum_{j=1}^{N} \mathrm{sgn}(\boldsymbol{x}_j^\top \boldsymbol{s})(\boldsymbol{x}_j^\top \boldsymbol{s}) \right\|_2^2 \tag{41}$$

$$= \left\| (\boldsymbol{I} - \boldsymbol{Q}\boldsymbol{Q}^\top) \sum_{j=1}^{N} \mathrm{sgn}(\boldsymbol{x}_j^\top \boldsymbol{b})\boldsymbol{x}_j \right\|_2^2 - \sin^2(\theta) \left( \sum_{j=1}^{N} |\boldsymbol{x}_j^\top \boldsymbol{s}| \right)^2 \tag{42}$$

Note that the above can be further written as

$$\left\| \sum_{j=1}^{N} \mathrm{sgn}(\boldsymbol{x}_j^\top \boldsymbol{b})\boldsymbol{x}_j \right\|_2^2 - \left\| \boldsymbol{Q}\boldsymbol{Q}^\top \sum_{j=1}^{N} \mathrm{sgn}(\boldsymbol{x}_j^\top \boldsymbol{b})\boldsymbol{x}_j \right\|_2^2 - \sin^2(\theta) \left( \sum_{j=1}^{N} |\boldsymbol{x}_j^\top \boldsymbol{s}| \right)^2 \tag{43}$$

Now, the first term can be lower bounded by

$$\left\| \sum_{j=1}^{N} \mathrm{sgn}(\boldsymbol{x}_j^\top \boldsymbol{b})\boldsymbol{x}_j \right\|_2^2 \stackrel{\|\boldsymbol{s}\|=1}{=} \left\| \sum_{j=1}^{N} \mathrm{sgn}(\boldsymbol{x}_j^\top \boldsymbol{b})\boldsymbol{x}_j \right\|_2^2 \|\boldsymbol{s}\|_2^2 \tag{44}$$

$$\stackrel{\text{Cauchy-Schwartz}}{\geq} \left( \sum_{j=1}^{N} \mathrm{sgn}(\boldsymbol{x}_j^\top \boldsymbol{b})\boldsymbol{x}_j^\top \boldsymbol{s} \right)^2 \tag{45}$$

$$\stackrel{(36)}{=} \left( \sum_{j=1}^{N} \mathrm{sgn}(\boldsymbol{x}_j^\top \boldsymbol{s})\boldsymbol{x}_j^\top \boldsymbol{s} \right)^2 = \left( \sum_{j=1}^{N} |\boldsymbol{x}_j^\top \boldsymbol{s}| \right)^2. \tag{46}$$

Substituting this back to (43), we have

$$\left\| (\boldsymbol{I} - \boldsymbol{b}\boldsymbol{b}^\top - \boldsymbol{Q}\boldsymbol{Q}^\top) \sum_{j=1}^{N} \mathrm{sgn}(\boldsymbol{x}_j^\top \boldsymbol{b})\boldsymbol{x}_j \right\|_2^2 \tag{47}$$

$$\geq \cos^2(\theta) \left( \sum_{j=1}^{N} |\boldsymbol{x}_j^\top \boldsymbol{s}| \right)^2 - \left\| \boldsymbol{Q}\boldsymbol{Q}^\top \sum_{j=1}^{N} \mathrm{sgn}(\boldsymbol{x}_j^\top \boldsymbol{b})\boldsymbol{x}_j \right\|_2^2. \tag{48}$$

This leads to final lower bound

$$\left\| (\boldsymbol{I} - \boldsymbol{b}\boldsymbol{b}^\top - \boldsymbol{Q}\boldsymbol{Q}^\top) \sum_{j=1}^{N} \mathrm{sgn}(\boldsymbol{x}_j^\top \boldsymbol{b})\boldsymbol{x}_j \right\|_2^2 \tag{49}$$

$$\geq \cos^2(\theta) N^2 c_{\mathrm{in}}^2 - N^2 \gamma_{\mathrm{in},\theta}^2 \tag{50}$$

Note that the lower bound is a decreasing function of $\theta$.

**Upper bounding the size of subgradient for outliers.** From Assumption 2, we have that $\boldsymbol{b}$ is orthogonal to at most $D - 1$ outliers. Therefore, we have

$$(\boldsymbol{I} - \boldsymbol{b}\boldsymbol{b}^\top - \boldsymbol{Q}\boldsymbol{Q}^\top) \left( \sum_{j=1}^{M} \mathrm{sgn}(\boldsymbol{o}_j^\top \boldsymbol{b})\boldsymbol{o}_j \right) \tag{51}$$

$$= (\boldsymbol{I} - \boldsymbol{b}\boldsymbol{b}^\top - \boldsymbol{Q}\boldsymbol{Q}^\top) \left( \sum_{j=1}^{M} \mathrm{sign}(\boldsymbol{o}_j^\top \boldsymbol{b})\boldsymbol{o}_j + \sum_{j=1}^{D-1} \lambda_j \boldsymbol{o}_{l_j} \right). \tag{52}$$

Here sign is the usual sign function, $\{\boldsymbol{o}_{l_j}\}_{j=1}^{D-1}$ are $D-1$ outliers and $\lambda_j \in [-1, 1]$ for $j = 1, \ldots, D-1$. Note further that

$$\left\| (\boldsymbol{I} - \boldsymbol{b}\boldsymbol{b}^\top - \boldsymbol{Q}\boldsymbol{Q}^\top) \sum_{j=1}^{D-1} \lambda_j \boldsymbol{o}_{l_j} \right\|_2 \leq \left\| \sum_{j=1}^{D-1} \lambda_j \boldsymbol{o}_{l_j} \right\|_2 \leq \sum_{j=1}^{D-1} \left\| \lambda_j \boldsymbol{o}_{l_j} \right\|_2 \leq D. \tag{53}$$

The last inequality makes use of the fact that $\boldsymbol{o}_j \in \mathbb{S}^{D-1}$. Therefore, we have

$$\left\| (\boldsymbol{I} - \boldsymbol{b}\boldsymbol{b}^\top - \boldsymbol{Q}\boldsymbol{Q}^\top) \left( \sum_{j=1}^{M} \mathrm{sign}(\boldsymbol{o}_j^\top \boldsymbol{b})\boldsymbol{o}_j + \sum_{j=1}^{D-1} \lambda_j \boldsymbol{o}_{l_j} \right) \right\|_2 \tag{54}$$

$$\overset{\text{Triangle Ineq.}}{\leq} \left\| (\boldsymbol{I} - \boldsymbol{b}\boldsymbol{b}^\top - \boldsymbol{Q}\boldsymbol{Q}^\top) \left( \sum_{j=1}^{M} \mathrm{sign}(\boldsymbol{o}_j^\top \boldsymbol{b})\boldsymbol{o}_j \right) \right\|_2 + D, \tag{55}$$

$$\leq M\bar{\eta}_{\mathrm{out}}. \tag{56}$$

**Putting everything together.** From (35) it is necessary that the size of subgradients for inliers and outliers must be equal

$$\left\| (\boldsymbol{I} - \boldsymbol{b}\boldsymbol{b}^\top - \boldsymbol{Q}\boldsymbol{Q}^\top) \left( \sum_{j=1}^{N} \mathrm{sgn}(\boldsymbol{x}_j^\top \boldsymbol{b})\boldsymbol{x}_j \right) \right\| = \left\| (\boldsymbol{I} - \boldsymbol{b}\boldsymbol{b}^\top - \boldsymbol{Q}\boldsymbol{Q}^\top) \left( \sum_{j=1}^{M} \mathrm{sgn}(\boldsymbol{o}_j^\top \boldsymbol{b})\boldsymbol{o}_j \right) \right\|. \tag{57}$$

This together with (50) and (56) yields

$$\cos^2(\theta) N^2 c_{\mathrm{in}}^2 - N^2 \gamma_{\mathrm{in},\theta}^2 \leq M^2 \bar{\eta}_{\mathrm{out}}^2. \tag{58}$$

Since $\cos^2(\theta) N^2 c_{\mathrm{in}}^2 - N^2 \gamma_{\mathrm{in},\theta}^2 - M^2 \bar{\eta}_{\mathrm{out}}^2$ is strictly decreasing in $\theta$, and (58) holds for $\theta = \frac{\pi}{2}$, there exists a unique $\theta_1 \geq 0$ such that (58) is equivalent to $\theta \geq \theta_1$. $\qquad \square$

**Theorem 3.** *Every global minimizer of* (OR$^2$-$\ell^1$) *has an angle away from* $\pm b^*$ *of at most*

$$\arcsin\left(\frac{M}{N} \cdot \frac{c_{\text{out,max}} - c_{\text{out,min}}}{c_{\text{in}}}\right). \tag{21}$$

*Proof.* Suppose $b$ is a global minimizer. If $b = \pm b^*$ then we are done, so suppose the otherwise. We can again decompose $b$ as in (36). Now, on the one hand,

$$f(b) = \sum_{j=1}^{N} |x_j^\top b| + \sum_{j=1}^{M} |o_j^\top b| \tag{59}$$

$$= \sin(\theta) \sum_{j=1}^{N} |x_j^\top s| + \sum_{j=1}^{M} |o_j^\top b| \tag{60}$$

$$\geq \sin(\theta) N c_{\text{in}} + M c_{\text{out,min}}. \tag{61}$$

On the other hand,

$$f(b) \leq f(b^*) = \sum_{j=1}^{M} |o_j^\top b^*| \leq M c_{\text{out,max}}. \tag{62}$$

Combining the above, it must hold that

$$M c_{\text{out,max}} \geq \sin(\theta) N c_{\text{in}} + M c_{\text{out,min}}, \tag{63}$$

and we conclude the proof. $\qquad\square$

Theorem 2 then follows from Lemma 6 and Theorem 3, and Theorem 1 is a result of Theorem 2. We conclude the section with the following remark. While the high-level proof strategy is motivated by Zhu et al. (2018), the problem we study (OR$^2$-$\ell^1$) handles *general* manifolds $\mathcal{M}$'s satisfying Assumption 1, while Zhu et al. (2018) considers only $\mathcal{M} = \mathbb{S}^{D-1}$. Further, our analysis induces novel quantities $c_{\text{in}}, \gamma_{\text{in},\theta}$ that are critical to tolerating non-uniform inliers, while Zhu et al. (2018) requires uniform inliers, as explained in §4.1.

### B.2  CONVERGENCE PROOF OF PSGM

It turns out to be useful to define for any $b_1, b_2 \in \mathbb{S}^{D-1}$

$$\operatorname{dist}(b_1, b_2) := \min_{\tau \in \{\pm 1\}} \|b_1 - \tau b_2\|_2 = \sqrt{2 - 2|\langle b_1, b_2\rangle|} = 2\sin(\angle(b_1, b_2)/2). \tag{64}$$

**Lemma 7** (Sharpness of (OR$^2$-$\ell^1$)). *The objective* $f(\cdot)$ *of* (OR$^2$-$\ell^1$) *satisfies*

$$f(b) - f(b^*) \geq s \operatorname{dist}(b, b^*), \quad \forall b \in \mathcal{M}, \tag{65}$$

*where* $s := \frac{1}{2} N c_{\text{in}} - M c_{\text{out,max}}$.

*Proof.* It is clear that when $b = \pm b^*$, (65) holds for any $s > 0$ so we are done. Suppose $b \neq \pm b^*$. Again taking a decomposition of $b$ as in (36), we have

$$f(b) - f(b^*) = \sin(\theta) \sum_{j=1}^{N} |x_j^\top s| + \sum_{j=1}^{M} |o_j^\top b| - \sum_{j=1}^{M} |o_j^\top b^*| \tag{66}$$

$$\geq \sin(\theta) N c_{\text{in}} + \sum_{j=1}^{M} (|o_j^\top b| - |o_j^\top b^*|). \tag{67}$$

To lower bound the first term, we observe that

$$\operatorname{dist}(b, b^*) = 2\sin(\theta/2) \leq 2\sin(\theta), \tag{68}$$

where the equality follows from (64), and the inequality is due to $\sin(\cdot)$ is increasing on $[0, \pi/2]$. Now we lower bound the last two terms in (67). For any $\tau \in \{\pm 1\}$, the following holds

$$\sum_{j=1}^{M} (|\boldsymbol{o}_j^\top \boldsymbol{b}| - |\boldsymbol{o}_j^\top \boldsymbol{b}^*|) = \sum_{j=1}^{M} (|\boldsymbol{o}_j^\top \tau \boldsymbol{b}| - |\boldsymbol{o}_j^\top \boldsymbol{b}^*|) \tag{69}$$

$$= \sum_{j=1}^{M} (|\boldsymbol{o}_j^\top (\tau \boldsymbol{b} - \boldsymbol{b}^* + \boldsymbol{b}^*)| - |\boldsymbol{o}_j^\top \boldsymbol{b}^*|) \tag{70}$$

$$\geq -\sum_{j=1}^{M} |\boldsymbol{o}_j^\top (\tau \boldsymbol{b} - \boldsymbol{b}^*)| \tag{71}$$

$$= -\|\tau \boldsymbol{b} - \boldsymbol{b}^*\|_2 \sum_{j=1}^{M} \left| \boldsymbol{o}_j^\top \frac{\tau \boldsymbol{b} - \boldsymbol{b}^*}{\|\tau \boldsymbol{b} - \boldsymbol{b}^*\|_2} \right| \tag{72}$$

$$\geq -\|\tau \boldsymbol{b} - \boldsymbol{b}^*\|_2 M c_{\text{out,max}} \tag{73}$$

where (71) is from the triangle inequality. Therefore, it is true in particular that

$$\sum_{j=1}^{M} (|\boldsymbol{o}_j^\top \boldsymbol{b}| - |\boldsymbol{o}_j^\top \boldsymbol{b}^*|) \geq -\operatorname{dist}(\boldsymbol{b}, \boldsymbol{b}^*) M c_{\text{out,max}}. \tag{74}$$

Combining (67), (68) and (74) yields

$$f(\boldsymbol{b}) - f(\boldsymbol{b}^*) \geq \left( \frac{1}{2} N c_{\text{in}} - M c_{\text{out,max}} \right) \operatorname{dist}(\boldsymbol{b}, \boldsymbol{b}^*). \tag{75}$$

$\square$

**Lemma 8** (Distance Lemma). *Let $\{\boldsymbol{b}_k\}_{k=0}^{K}$ be the sequence generated by Algorithm 1. Let $\kappa$ be a Lipschitz constant for $f(\cdot)$ restricted to $\mathcal{M}$. Then, it holds for any $\boldsymbol{b} \in \mathcal{M}$ that*

$$\|\boldsymbol{b}_k - \boldsymbol{b}\|_2^2 \leq \|\boldsymbol{b}_{k-1} - \boldsymbol{b}\|_2^2 + (c_1 + 1)\alpha_{k-1}^2 \kappa^2 + 2\alpha_{k-1}(f(\boldsymbol{b}) - f(\boldsymbol{b}_{k-1})) + 2\alpha_{k-1} c_2 \|\boldsymbol{b}_{k-1} - \boldsymbol{b}\|_2^2, \tag{76}$$

*where $c_2 = \kappa \cdot c_3$, and $c_1, c_3$ are positive constants that depend only on $\mathcal{M}$.*

*Proof.* Since $f(\cdot)$ is convex, it follows from (Li et al., 2019, Corollary 1) that

$$f(\boldsymbol{b}) - f(\boldsymbol{b}_{k-1}) + c_2 \|\boldsymbol{b}_{k-1} - \boldsymbol{b}\|_2^2 \geq \langle \boldsymbol{b} - \boldsymbol{b}_{k-1}, \boldsymbol{r}_{k-1} \rangle. \tag{77}$$

Therefore, we have

$$\|\boldsymbol{b}_k - \boldsymbol{b}\|_2^2 = \left\| \operatorname{retr}_{\boldsymbol{b}_{k-1}}(-\alpha_{k-1}\boldsymbol{r}_{k-1}) - \boldsymbol{b} \right\|_2^2 \tag{78}$$

$$\leq \|\boldsymbol{b}_{k-1} - \alpha_{k-1}\boldsymbol{r}_{k-1} - \boldsymbol{b}\|_2^2 + c_1 \|\alpha_{k-1}\boldsymbol{r}_{k-1}\|_2^2 \tag{79}$$

$$= \|\boldsymbol{b}_{k-1} - \boldsymbol{b}\|_2^2 + (c_1 + 1)\|\alpha_{k-1}\boldsymbol{r}_{k-1}\|_2^2 + 2\langle \boldsymbol{b} - \boldsymbol{b}_{k-1}, \alpha_{k-1}\boldsymbol{r}_{k-1} \rangle \tag{80}$$

$$\leq \|\boldsymbol{b}_{k-1} - \boldsymbol{b}\|_2^2 + (c_1 + 1)\alpha_{k-1}^2 \kappa^2 + 2\alpha_{k-1}\langle \boldsymbol{b} - \boldsymbol{b}_{k-1}, \boldsymbol{r}_{k-1} \rangle \tag{81}$$

$$\leq \|\boldsymbol{b}_{k-1} - \boldsymbol{b}\|_2^2 + (c_1 + 1)\alpha_{k-1}^2 \kappa^2 + 2\alpha_{k-1}(f(\boldsymbol{b}) - f(\boldsymbol{b}_{k-1})) + 2\alpha_{k-1} c_2 \|\boldsymbol{b}_{k-1} - \boldsymbol{b}\|_2^2,$$

where the first inequality follows from Assumption 1 and (Boumal et al., 2019, Appendix B), $c_1$ is a constant that depends only on $\mathcal{M}$; the second inequality follows from the Lipschitzness of $f(\cdot)$, and the last inequality follows from the argument above. $\square$

**Theorem 4.** *With a suitable initialization and geometrically diminishing step size, the sequence $\{\boldsymbol{b}_k\}_{k=0}^{K}$ given by Algorithm 1 converges at a linear rate to the true vector $\pm \boldsymbol{b}^*$. More precisely, with the step size $\alpha_k = \beta^k \cdot \alpha_0$, the principal angle $\theta_k$ between $\boldsymbol{b}_k$ and $\boldsymbol{b}^*$ is upper bounded as per*

$$\sin(\theta_k/2) \leq \beta^k \sin(\theta_0/2) \tag{22}$$

*if the following conditions are satisfied:*

(i) *Initialization not too far from $\pm \boldsymbol{b}^*$:* $\sin(\theta_0/2) < \frac{s}{2c_2}$,

(ii) *Small initial step size:* $0 < \alpha_0 < \min\left(\frac{\sin(\theta_0/2)}{s - 2c_2 \sin(\theta_0/2)}, \frac{4(s - 2c_2 \sin(\theta_0/2))}{(c_1+1)\kappa^2/\sin(\theta_0/2)}\right)$,

(iii) *Suitable step size decay:* $1 > \beta \geq \sqrt{1 + \alpha_0\left(\alpha_0 \frac{(c_1+1)\kappa^2}{4\sin(\theta_0/2)^2} + 2c_2 - \frac{s}{\sin(\theta_0/2)}\right)}$.

*In the above, $s := \frac{1}{2}Nc_{\text{in}} - Mc_{\text{out,max}}$, $\kappa$ is a Lipschitz constant of the objective $f(\cdot)$, $c_2 = \kappa \cdot c_3$, and $c_1$ and $c_3$ are constants that depend only on $\mathcal{M}$.*

*Proof.* Due to (64), it suffices to prove that

$$\text{dist}(\boldsymbol{b}_k, \boldsymbol{b}^*) \leq \beta^k \text{dist}(\boldsymbol{b}_0, \boldsymbol{b}^*). \tag{82}$$

We prove the theorem by induction on $k$. When $k = 0$, (22) holds trivially. Suppose $k > 0$, and let $\tau^* := \text{argmin}_{\tau \in \{\pm 1\}} \|\boldsymbol{b}_{k-1} - \tau \boldsymbol{b}^*\|_2^2$. Then,

$$\text{dist}^2(\boldsymbol{b}_k, \boldsymbol{b}^*) = \min_{\tau \in \{\pm 1\}} \|\boldsymbol{b}_k - \tau \boldsymbol{b}^*\|_2^2 \leq \|\boldsymbol{b}_k - \tau^* \boldsymbol{b}^*\|_2^2 \tag{83}$$

$$\leq (1 + 2\alpha_{k-1}c_2)\|\boldsymbol{b}_{k-1} - \tau^* \boldsymbol{b}^*\|_2^2 + (c_1 + 1)\alpha_{k-1}^2\kappa^2 + 2\alpha_{k-1}(f(\boldsymbol{b}^*) - f(\boldsymbol{b}_{k-1})) \tag{84}$$

$$= (1 + 2\alpha_{k-1}c_2)\text{dist}^2(\boldsymbol{b}_{k-1}, \boldsymbol{b}^*) + (c_1 + 1)\alpha_{k-1}^2\kappa^2 + 2\alpha_{k-1}(f(\boldsymbol{b}^*) - f(\boldsymbol{b}_{k-1})) \tag{85}$$

$$\leq (1 + 2\alpha_0 c_2)\text{dist}^2(\boldsymbol{b}_{k-1}, \boldsymbol{b}^*) - 2\alpha_{k-1}s\, \text{dist}(\boldsymbol{b}_{k-1}, \boldsymbol{b}^*) + (c_1 + 1)\alpha_{k-1}^2\kappa^2. \tag{86}$$

To upper bound it Consider the quadratic function $l(t) := (1 + 2\alpha_0 c_2)t^2 - 2\alpha_{k-1}st + (c_1+1)\alpha_{k-1}^2\kappa^2$. It has an axis of symmetry given by the vertical line $t = t_0$, where $t_0 := \frac{\alpha_{k-1}s}{1 + 2\alpha_0 c_2} > 0$; so $l(0) = l(2t_0)$. On the other hand, since $\alpha_0 < \frac{\sin(\theta_0/2)}{s - 2c_2 \sin(\theta_0/2)}$ and $s - 2c_2 \sin(\theta_0/2) > 0$ due to conditions (i) and (ii), we have

$$\alpha_0(s - 2c_2 \sin(\theta_0/2)) < \sin(\theta_0/2) \tag{87}$$

which by (64) is equivalent to

$$2\alpha_0(s - c_2 \text{dist}(\boldsymbol{b}_0, \boldsymbol{b}^*)) < \text{dist}(\boldsymbol{b}_0, \boldsymbol{b}^*) \tag{88}$$

$$\Leftrightarrow \frac{2\alpha_0 s}{1 + 2\alpha_0 c_2} < \text{dist}(\boldsymbol{b}_0, \boldsymbol{b}^*) \tag{89}$$

$$\Leftrightarrow \frac{2\alpha_0 \beta^{k-1} s}{1 + 2\alpha_0 c_2} < \beta^{k-1} \text{dist}(\boldsymbol{b}_0, \boldsymbol{b}^*). \tag{90}$$

This is to say that $2t_0 < \beta^{k-1} \text{dist}(\boldsymbol{b}_0, \boldsymbol{b}^*)$. Since the induction hypothesis is

$$\text{dist}(\boldsymbol{b}_{k-1}, \boldsymbol{b}^*) \leq \beta^{k-1} \text{dist}(\boldsymbol{b}_0, \boldsymbol{b}^*), \tag{91}$$

we have $l(\text{dist}(\boldsymbol{b}_{k-1}, \boldsymbol{b}^*)) < l(\beta^{k-1} \text{dist}(\boldsymbol{b}_0, \boldsymbol{b}^*))$. Combining this with (86) yields

$$\text{dist}^2(\boldsymbol{b}_k, \boldsymbol{b}^*) \leq (1 + 2\alpha_0 c_2)\beta^{2k-2} \text{dist}^2(\boldsymbol{b}_0, \boldsymbol{b}^*) - 2\beta^{2k-2}\alpha_0 s\, \text{dist}(\boldsymbol{b}_0, \boldsymbol{b}^*) + (c_1 + 1)\beta^{2k-2}\alpha_0^2\kappa^2$$

$$= \beta^{2k-2} \text{dist}^2(\boldsymbol{b}_0, \boldsymbol{b}^*)\left(1 + 2\alpha_0 c_2 - 2\frac{\alpha_0 s}{\text{dist}(\boldsymbol{b}_0, \boldsymbol{b}^*)} + \frac{(c_1 + 1)\alpha_0^2\kappa^2}{\text{dist}^2(\boldsymbol{b}_0, \boldsymbol{b}^*)}\right) \tag{92}$$

$$\leq \beta^{2k} \text{dist}^2(\boldsymbol{b}_0, \boldsymbol{b}^*), \tag{93}$$

where the last inequality follows from condition (iii). This completes the induction step. Finally, note that with condition (i) in place, (ii) is not vacuous since both terms in $\min$ are positive; thus (iii) is also not vacuous, since (ii) implies $\alpha_0 \frac{(c_1+1)\kappa^2}{4\sin(\theta_0/2)^2} + 2c_2 - \frac{s}{\sin(\theta_0/2)} < 0$. $\square$

## B.3 CONVERGENCE PROOF OF IRLS

**Lemma 9.** *The weighted eigenvalue subproblem (10) in Algorithm 2 is equivalent to*

$$\boldsymbol{b}_{k+1} \in \text{argmin}_{\boldsymbol{b} \in \mathcal{M}} \sum_{j=1}^{L} g(\boldsymbol{y}_j^\top \boldsymbol{b}, \boldsymbol{y}_j^\top \boldsymbol{b}_k), \qquad g(r, v) := h(v) + \frac{r^2 - v^2}{2 \cdot \max\{|v|, \delta\}}. \tag{94}$$

*Furthermore, $g$ is a tight upper bound of $h$ of ($\mathrm{OR}^2$-Huber): For every $r \in \mathbb{R}$ and every $v \in \mathbb{R}$ we have*

$$h(r) \leq g(r, v)$$
$$h'(r) = \frac{\partial g(r, v)}{\partial r}\bigg|_{r=v}$$

*In the first inequality, equality is attained if and only if $r = v$.*

*Proof.* This is an elementary result, and the proof is omitted. □

**Lemma 10** (Lipschitz Smoothness). *Recall the definition of $g(\cdot, \cdot)$ in (94). For every $\boldsymbol{b} \in \mathcal{M}$, every $\boldsymbol{b}' \in \mathcal{M}$, and every $\boldsymbol{z} \in \mathcal{M}$ we have*

$$\left\|\nabla g(\boldsymbol{y}_j^\top \boldsymbol{b}, \boldsymbol{y}_j^\top \boldsymbol{z}) - \nabla g(\boldsymbol{y}_j^\top \boldsymbol{b}', \boldsymbol{y}_j^\top \boldsymbol{z})\right\|_2 \leq \frac{1}{\delta}\|\boldsymbol{y}_j\|_2^2 \cdot \left\|\boldsymbol{b} - \boldsymbol{b}'\right\|_2$$

*In the above, $\nabla g(\boldsymbol{y}_j^\top \boldsymbol{b}, \boldsymbol{y}_j^\top \boldsymbol{z})$ denotes the gradient of $g$ with respect to $\boldsymbol{b}$.*

*Proof.* First note that for each $\boldsymbol{b} \in \mathcal{M}$ and $\boldsymbol{z} \in \mathcal{M}$ we have

$$\nabla g(\boldsymbol{y}_j^\top \boldsymbol{b}, \boldsymbol{y}_j^\top \boldsymbol{z}) = \frac{\boldsymbol{y}_j \boldsymbol{y}_j^\top \boldsymbol{b}}{\max\{|\boldsymbol{y}_j^\top \boldsymbol{z}|, \delta\}}$$

Therefore

$$
\begin{aligned}
\left\|\nabla g(\boldsymbol{y}_j^\top \boldsymbol{b}, \boldsymbol{y}_j^\top \boldsymbol{z}) - \nabla g(\boldsymbol{y}_j^\top \boldsymbol{b}', \boldsymbol{y}_j^\top \boldsymbol{z})\right\|_2 &= \left\|\frac{\boldsymbol{y}_j \boldsymbol{y}_j^\top \boldsymbol{b}}{\max\{|\boldsymbol{y}_j^\top \boldsymbol{z}|, \delta\}} - \frac{\boldsymbol{y}_j \boldsymbol{y}_j^\top \boldsymbol{b}'}{\max\{|\boldsymbol{y}_j^\top \boldsymbol{z}|, \delta\}}\right\|_2 \\
&\leq \left\|\frac{\boldsymbol{y}_j \boldsymbol{y}_j^\top}{\max\{|\boldsymbol{y}_j^\top \boldsymbol{z}|, \delta\}}\right\|_2 \cdot \left\|\boldsymbol{b} - \boldsymbol{b}'\right\|_2 \\
&\leq \frac{1}{\delta}\|\boldsymbol{y}_j\|_2^2 \cdot \left\|\boldsymbol{b} - \boldsymbol{b}'\right\|_2
\end{aligned}
$$

The proof is finished. □

**Lemma 11** (Sufficient Decrease). *The sequence $\{\boldsymbol{b}_k\}_{k=0}^K$ produced by Algorithm 2 satisfies*

$$\sum_{j=1}^L g(\boldsymbol{y}_j^\top \boldsymbol{b}_{k+1}, \boldsymbol{y}_j^\top \boldsymbol{b}_k) - \sum_{j=1}^L g(\boldsymbol{y}_j^\top \boldsymbol{b}_k, \boldsymbol{y}_j^\top \boldsymbol{b}_k) \leq -\frac{1}{2\sigma}\left\|\mathrm{rgrad}\sum_{j=1}^L g(\boldsymbol{y}_j^\top \boldsymbol{b}_k, \boldsymbol{y}_j^\top \boldsymbol{b}_k)\right\|_2^2.$$

*Here, $\sigma$ is a finite positive constant depending on the manifold $\mathcal{M}$ and data $\boldsymbol{y}_j$'s.*

*Proof.* By Lemma 10 we know that $\sum_{j=1}^L g(\boldsymbol{y}_j^\top \boldsymbol{b}, \boldsymbol{y}_j^\top \boldsymbol{b}_k)$ is Lipschitz smooth in $\boldsymbol{b}$ (with constant $\frac{1}{\delta}\sum_{j=1}^L \|\boldsymbol{y}_j\|_2^2$). Note also that $\mathcal{M}$ is a compact smooth submanifold of $\mathbb{S}^{D-1}$ by Assumption 1. Therefore, it follows from Appendix B of Boumal et al. (2019) that there is some finite constant $\sigma > 0$ satisfying

$$\sum_{j=1}^L g(\boldsymbol{y}_j^\top \mathrm{retr}_{\boldsymbol{b}_k}(\boldsymbol{r}), \boldsymbol{y}_j^\top \boldsymbol{b}_k) \leq \sum_{j=1}^L g(\boldsymbol{y}_j^\top \boldsymbol{b}_k, \boldsymbol{y}_j^\top \boldsymbol{b}_k) + \boldsymbol{r}^\top\left(\mathrm{rgrad}\sum_{j=1}^L g(\boldsymbol{y}_j^\top \boldsymbol{b}_k, \boldsymbol{y}_j^\top \boldsymbol{b}_k)\right) + \frac{\sigma}{2}\|\boldsymbol{r}\|_2^2$$

for every $\boldsymbol{r} \in T_{\boldsymbol{b}_k}\mathcal{M}$; $\sigma$ depends on the manifold $\mathcal{M}$ and the Lipschitz smoothness $\frac{1}{\delta}\sum_{j=1}^L \|\boldsymbol{y}_j\|_2^2$ of $\sum_{j=1}^L g(\boldsymbol{y}_j^\top \boldsymbol{b}, \boldsymbol{y}_j^\top \boldsymbol{b}_k)$, i.e., $\sigma$ depends on $\mathcal{M}$, data $\boldsymbol{y}_j$'s, and $\delta$. In particular, the above holds with $\boldsymbol{r} = -\frac{1}{\sigma}\mathrm{rgrad}\sum_{j=1}^L g(\boldsymbol{y}_j^\top \boldsymbol{b}_k, \boldsymbol{y}_j^\top \boldsymbol{b}_k)$, meaning that

$$\sum_{j=1}^L g(\boldsymbol{y}_j^\top \mathrm{retr}_{\boldsymbol{b}_k}(\boldsymbol{r}), \boldsymbol{y}_j^\top \boldsymbol{b}_k) \leq \sum_{j=1}^L g(\boldsymbol{y}_j^\top \boldsymbol{b}_k, \boldsymbol{y}_j^\top \boldsymbol{b}_k) - \frac{1}{2\sigma}\left\|\mathrm{rgrad}\sum_{j=1}^L g(\boldsymbol{y}_j^\top \boldsymbol{b}_k, \boldsymbol{y}_j^\top \boldsymbol{b}_k)\right\|_2^2.$$

Since $\boldsymbol{b}_{k+1}$ is a global minimizer of $\sum_{j=1}^L g(\boldsymbol{y}_j^\top \boldsymbol{b}, \boldsymbol{y}_j^\top \boldsymbol{b}_k)$ in $\boldsymbol{b} \in \mathcal{M}$, $\sum_{j=1}^L g(\boldsymbol{y}_j^\top \boldsymbol{b}_{k+1}, \boldsymbol{y}_j^\top \boldsymbol{b}_k)$ is smaller than or equal to $\sum_{j=1}^L g(\boldsymbol{y}_j^\top \mathrm{retr}_{\boldsymbol{b}_k}(\boldsymbol{r}), \boldsymbol{y}_j^\top \boldsymbol{b}_k)$, and the proof is complete. □

**Theorem 5.** *The sequence $\{\boldsymbol{b}_k\}_{k=0}^K$ produced by Algorithm 2 converges at a sublinear rate to a critical point of the following problem:*

$$\min_{\boldsymbol{b}\in\mathcal{M}} \sum_{j=1}^{L} h(\boldsymbol{y}_j^\top \boldsymbol{b}), \qquad h(r) := \begin{cases} |r| & |r| > \delta \\ \frac{1}{2}\left(r^2/\delta + \delta\right) & |r| \leq \delta \end{cases} \qquad \text{(OR}^2\text{-Huber)}$$

*More precisely, the Riemannian gradient of the objective of (OR$^2$-Huber) is bounded above:*

$$\min_{k=0,\dots,K} \left\| \operatorname{rgrad} \sum_{j=1}^{L} h(\boldsymbol{y}_j^\top \boldsymbol{b}_k) \right\|_2 \leq \sqrt{\frac{2\sigma \cdot \sum_{j=1}^{L} h(\boldsymbol{y}_j^\top \boldsymbol{b}_0)}{K+1}} \qquad (23)$$

*In (23), $\sigma$ is a finite positive constant depending on the manifold $\mathcal{M}$, data $\boldsymbol{y}_j$'s, and $\delta$.*

*Proof.* For every $k = 0, \dots, K$ we have

$$\sum_{j=1}^{L} h(\boldsymbol{y}_j^\top \boldsymbol{b}_{k+1}) - \sum_{j=1}^{L} h(\boldsymbol{y}_j^\top \boldsymbol{b}_k) \leq \sum_{j=1}^{L} g\big(\boldsymbol{y}_j^\top \boldsymbol{b}_{k+1}, \boldsymbol{y}_j^\top \boldsymbol{b}_k\big) - \sum_{j=1}^{L} g\big(\boldsymbol{y}_j^\top \boldsymbol{b}_k, \boldsymbol{y}_j^\top \boldsymbol{b}_k\big)$$

$$\leq -\frac{1}{2\sigma} \left\| \operatorname{rgrad} \sum_{j=1}^{L} g\big(\boldsymbol{y}_j^\top \boldsymbol{b}_k, \boldsymbol{y}_j^\top \boldsymbol{b}_k\big) \right\|_2^2$$

where the first inequality follows from Lemma 9 and the second from Lemma 11. Rearrange the above inequality and sum over $k = 0, \dots, K$ and we obtain

$$\sum_{k=0}^{K} \left\| \operatorname{rgrad} \sum_{j=1}^{L} g\big(\boldsymbol{y}_j^\top \boldsymbol{b}_k, \boldsymbol{y}_j^\top \boldsymbol{b}_k\big) \right\|_2^2 \leq 2\sigma \Big( \sum_{j=1}^{L} h(\boldsymbol{y}_j^\top \boldsymbol{b}_0) - \sum_{j=1}^{L} h(\boldsymbol{y}_j^\top \boldsymbol{b}_{K+1}) \Big)$$

Finally, by Lemma 9 we have $\nabla \sum_{j=1}^{L} g\big(\boldsymbol{y}_j^\top \boldsymbol{b}_k, \boldsymbol{y}_j^\top \boldsymbol{b}_k\big) = \nabla \sum_{j=1}^{L} h(\boldsymbol{y}_j^\top \boldsymbol{b})$, and therefore

$$\operatorname{rgrad} \sum_{j=1}^{L} g\big(\boldsymbol{y}_j^\top \boldsymbol{b}_k, \boldsymbol{y}_j^\top \boldsymbol{b}_k\big) = \operatorname{rgrad} \sum_{j=1}^{L} h(\boldsymbol{y}_j^\top \boldsymbol{b}).$$

Combining the above yields

$$\sum_{k=0}^{K} \left\| \operatorname{rgrad} \sum_{j=1}^{L} h(\boldsymbol{y}_j^\top \boldsymbol{b}) \right\|_2^2 \leq 2\sigma \Big( \sum_{j=1}^{L} h(\boldsymbol{y}_j^\top \boldsymbol{b}_0) - \sum_{j=1}^{L} h(\boldsymbol{y}_j^\top \boldsymbol{b}_{K+1}) \Big),$$

which means

$$(K+1) \cdot \min_{k=0,\dots,K} \left\| \operatorname{rgrad} \sum_{j=1}^{L} h(\boldsymbol{y}_j^\top \boldsymbol{b}) \right\|_2^2 \leq 2\sigma \cdot \Big( \sum_{j=1}^{L} h(\boldsymbol{y}_j^\top \boldsymbol{b}_0) - \sum_{j=1}^{L} h(\boldsymbol{y}_j^\top \boldsymbol{b}_{K+1}) \Big),$$

$$\leq 2\sigma \cdot \sum_{j=1}^{L} h(\boldsymbol{y}_j^\top \boldsymbol{b}_0)$$

concluding the proof. $\qquad\square$