# OpenReview forum: "Outlier-Robust Orthogonal Regression on Manifolds"
_ICLR.cc/2024/Conference — Submitted to ICLR 2024_

### Official Review · Reviewer_CDEb · 2023-10-27

**Soundness:** 2 fair
**Presentation:** 2 fair
**Contribution:** 2 fair
**Rating:** 3
**Confidence:** 4

**Summary:**

The authors formulate a way to do orthogonal regression on the sphere. The authors have theory on convergence rate of their estimators and on the geometry of the outliers/inliers.

**Strengths:**

The paper did a nice job of proposing a lot of theoretical quantities that could be useful for this type of research.

**Weaknesses:**

The direction of the paper feels unclear. At times the paper feels like it is directed towards general manifolds (as the title implies) but then Assumption 1 restricts to the case where M is a smooth compact submanifold of a sphere. This leads me to my reservation with the use of Algorithm 1. Algorithm 1 has a general setting to it with retractions and projections, however for spheres these maps are well defined in closed form (the log in exponential maps) which could perform better than the general mappings. The first line of the paper says they propose OR^2-l0 but their contribution is in terms of the l1 norm. It feels like this was a much longer paper or two papers that split at the seams.

There is a good amount of literature on manifold regression such as that of Fletcher et al. which I cannot seem to consolidate with the current work. For instance, on sphere it is well known that having antipodal sets of points causes issues even with mean estimation so I wish some sort of discussion to that effect were included.

When presenting the results I'm curious why only the median result is presented. Error bars would be beneficial.

There is too large an emphasis on theoretical results and very little demonstration of the theory.

**Questions:**

I have included some of my questions in the weaknesses.

---

### Official Review · Reviewer_ovBv · 2023-10-31

**Soundness:** 2 fair
**Presentation:** 3 good
**Contribution:** 2 fair
**Rating:** 3
**Confidence:** 4

**Summary:**

This paper considered the problem of outlier-robust orthogonal regression on a smooth and compact submanifold $\mathcal{M}$ of the unit sphere $\mathbb{S}^{D-1}$, i.e., finding a point in $\mathcal{M}$ that satisfies as many linear equations as possible. The authors provided some conditions on the geometry of inliers, outliers, the manifold $\mathcal{M}$ that make the minimizer of $OR^2$-$l^1$ can recover or is close to the ground truth. Two algorithms, i.e., Riemannian subgradient method and iteratively reweighted least squares method, were proposed to solve the $OR^2$-$l^1$ problem. Furthermore, the authors showed that with suitable initialization and step size, the Riemannian subgradient method converges to the ground truth; the iteratively reweighted least squares converges sub-linearly to a critical point of a surrogate $OR^2$-Huber.

**Strengths:**

This paper studied a very interesting problem of outlier-robust regression in the community of computer vision and machine learning. The core contribution is that the authors presented some assumptions and conditions (though not computable in practice) and showed the convergence of Riemannian subgradient method and iteratively reweighted least squares. The proofs did require some effort, unfortunately...

**Weaknesses:**

Unfortunately, the authors made some (possible serious) mistakes in proving their theorems.

(1) Page 17, (39) (40): $\sin(\theta)$ is missing in the inner product.

(2) Page 17, (43) is wrong (a serious mistake), which makes (48), (50) and even the whole theorem in doubt.
\begin{align*}
\left\Vert(\mathbf{I}-\mathbf{Q}\mathbf{Q}^T)\sum\_{j=1}^N {\rm{sgn}}(\mathbf{x}\_j^T\mathbf{b})\mathbf{x}\_j\right\Vert^2&=\left\Vert\sum\_{j=1}^N {\rm{sgn}}(\mathbf{x}\_j^T\mathbf{b})\mathbf{x}\_j\right\Vert^2+\left\Vert\mathbf{Q}\mathbf{Q}^T\sum\_{j=1}^N {\rm{sgn}}(\mathbf{x}\_j^T\mathbf{b})\mathbf{x}\_j\right\Vert^2\\\\
&-2\left\langle\sum\_{j=1}^N {\rm{sgn}}(\mathbf{x}\_j^T\mathbf{b})\mathbf{x}\_j,\mathbf{Q}\mathbf{Q}^T\sum\_{j=1}^N {\rm{sgn}}(\mathbf{x}\_j^T\mathbf{b})\mathbf{x}\_j\right\rangle\\\\
&\neq \left\Vert\sum\_{j=1}^N {\rm{sgn}}(\mathbf{x}\_j^T\mathbf{b})\mathbf{x}\_j\right\Vert^2-\left\Vert\mathbf{Q}\mathbf{Q}^T\sum\_{j=1}^N {\rm{sgn}}(\mathbf{x}\_j^T\mathbf{b})\mathbf{x}\_j\right\Vert^2
\end{align*}

(3) Page 18, (52) is in doubt. Suppose the authors abused the notation "sgn" and "sign", then the authors in fact added a zero term $(\mathbf{I}-\mathbf{b}\mathbf{b}^T-\mathbf{Q}\mathbf{Q}^T)\left(\sum_{j=1}^{D-1}\lambda_j\mathbf{o}_{l_j}\right)$ into (51). The funny thing is that the authors spent some effort to show this zero term is $\leq D$ (a non-zero constant).

(4) Page 19 (same as Page 7), (21) is nonsense if $\frac{M}{N}\cdot\frac{c_{out,max}-c_{out,min}}{c_{in}}>1$. This is possible if we have much more outliers than inliers, i.e., $M\gg N$.

(5) Page 19, the reviewer cannot see why "Theorem 2 then follows from Lemma 6 and Theorem 3, and Theorem 1 is a result of Theorem 2", please show them. Particularly, both Lemma 6 and Theorem 3 are in doubt.

(6) Page 19, first line of the proof of Lemma 7: If $\mathbf{b}=-\mathbf{b}^*$, then ${\rm{dist}}(\mathbf{b},\mathbf{b}^*)=2$. Why does (65) hold for any $s>0$?

(7) Page 23, please explain why $\nabla\sum\_{j=1}^Lg(\mathbf{y}\_j^T\mathbf{b}\_k,\mathbf{y}\_j^T\mathbf{b}\_k)=\nabla\sum\_{j=1}^Lh(\mathbf{y}\_j^T\mathbf{b})$? What is $\mathbf{b}$?

(8) Page 23, it seems the authors haven't proved why $\\{\mathbf{b}\_k\\}\_{k=0}^K$ converges at a sublinear rate to a critical point of the $OR^2$-Huber problem.

**Questions:**

1. What the colorbars represent in Figures 1, 2 and 3?

2. Page 4, Remark 2: For outliers from a uniform distribution over $\mathbb{S}^{D-1}$, there is no sense to discuss their linear independence because they are manifold-valued data. Shouldn't Assumption 2 be "Any size $\geq D$ subset of $\mathcal{O}$ are linearly independent"?

3. Page 5, Algorithm 1: Although the authors mentioned later that the step size $\alpha_k=\beta_k\cdot\alpha_0$, it is related to the ground truth. How to choose it in the experiments (cannot find it in Section 5 or Appendix A)?

4. Page 5, Algorithm 2: Similar as point 3, how to choose the parameter $\delta$ in practice?

5. Page 6, Figure 2: How to see that $c_{in}$ is large in the middle while it is small in the right? What are the exact numbers?

6. Page 6, first line of Remark 5: b should be bold.

7. Page 7, (19): cannot understand why it holds partly because the reviewer cannot find the proof of Theorem 2.

---

### Official Review · Reviewer_BNmQ · 2023-10-31

**Soundness:** 3 good
**Presentation:** 3 good
**Contribution:** 3 good
**Rating:** 3
**Confidence:** 3

**Summary:**

This paper is motivated by the problem of robust essential matrix estimation. For this purpose, it utilizes a known l1 optimization problem, but formulates it over a submanifold of the sphere, instead of previous works that studied it either on the sphere or the Grassmannian manifold.


The paper is clearly written and seems to be correct, however, it suffers from the following weaknesses:


1. The general motivating framework of the paper is the problem of robust subspace recovery and it had many relevant developments. In particular, the following works are not mentioned at all
Lerman, G., McCoy, M. B., Tropp, J. A., & Zhang, T. (2015). Robust computation of linear models by convex relaxation. Foundations of Computational Mathematics, 15, 363-410
Zhang, T. (2016). Robust subspace recovery by Tyler's M-estimator. Information and Inference: A Journal of the IMA, 5(1), 1-21
Zhang, T., & Lerman, G. (2014). A novel m-estimator for robust pca. The Journal of Machine Learning Research, 15(1), 749-808
Lerman, G., & Maunu, T. (2017). Fast, robust and non-convex subspace recovery. Information and Inference: A Journal of the IMA, 7(2), 277-336
Maunu, T., Zhang, T., & Lerman, G. (2019). A well-tempered landscape for non-convex robust subspace recovery. Journal of Machine Learning Research, 20(37)
Lerman G. &  Maunu T. (2018), "An Overview of Robust Subspace Recovery," in Proceedings of the IEEE, vol. 106, no. 8, pp. 1380-1410, Aug. 2018, doi: 10.1109/JPROC.2018.2853141
There is a lot of discussion of DPCP, however, this is just a framework of using the orthogonal component, instead of searching the subspace itself. In fact the main two practical methods of DPCP (gradient descent and IRLS) directly follow previous works of robust subspace recovery (gradient descent follows reference 5 above and IRLS follows reference 4 above), but instead of applying them to find a subspace, they find the orthogonal complement of this subspace and easily adapt the original methods. Here the orthogonal complement has dimension one, so the optimization is restricted to the sphere.
2. Many of the technical details follow previous works. Even though there is a new component of analyzing submanifolds of the sphere, the ideas are very similar to previous methods.  The authors mention some previous works of DPCP, but there are earlier works. For the non-convex, non-smooth setting, the well-tempered landscape and the geometric quantities for the geometry of the inliers and outliers and their interplay (permeance, alignment and stability) are discussed in reference 5 above (see https://arxiv.org/abs/1706.03896). A comparative discussion of the new ideas of the proposed work with respect to the developments of the latter work would be beneficial. Let me also mention that earlier ideas for these geometric quantities for a convex relaxation of this problem appeared in reference 1 above.
3. It is unclear exactly how many outliers in a realistic setting can be tolerated by the method. The thresholds presented in the theorem have no interpretation for any model of data, whether probabilistic or arbitrary.
4. The presentation mentions general manifolds, but it seems to me that the only possible application of this setting is to the manifold of essential matrices.
5. The experiments are rather weak in my opinion. I find it important to work on real computer vision datasets in order to witness the improvement gained by this method, while comparing to state of the art (SOTA) methods of robust estimation of essential matrices (as well as fundamental matrices, since the calibration is known in these datasets). Even the artificial experiments are unsatisfactory to me, since it is not clear how the outliers are generated and whether this generation corresponds to real scenarios, and there are no comparisons with other SOTA methods for robustly estimating the essential matrix.
6. I expect possibly stronger numerical results when working with fundamental matrices instead of essential matrices, since the corresponding set is higher-dimensional and in this sense it is easier to work with; however, I can see some other difficulties working with fundamental matrices using the proposed framework. It will be good to discuss this issue.

**Strengths:**

The paper is well written and motivated, and technically sound.

**Weaknesses:**

See comment 2: While the paper is technically rigorous and well-structured, it may appear to be an extension of existing work on robust subspace recovery at https://arxiv.org/abs/1706.03896. A comparative discussion with and against the existing work would enhance its contextual positioning.

**Questions:**

See comments 2 and 3 in summary.

---

### Official Review · Reviewer_6ASC · 2023-11-01

**Soundness:** 3 good
**Presentation:** 2 fair
**Contribution:** 2 fair
**Rating:** 5
**Confidence:** 3

**Summary:**

This work presented a generalized framework to L1 orthogonal regression problem as outlier-robust orthogonal regression over manifolds. 2 practical algorithms, OR2-RSGM and OR2-IRLS, are proposed under the OR2-L1 framework. Also, theoretical analysis and convergence rate proof are performed to support this work.

Empirical results for proposed method are evaluated on several synthetic data with essential matrix estimation problem, and it showed OR2-L1 algorithms achieved smaller error metric under relatively large outlier ratio when compared with previous work DPCP [1].

**Strengths:**

The core idea of this OR2-L1 framework as to go beyond unit spheres and change the feasible space in the optimization to be smooth and compact submanifold of the unit sphere, look reasonable from high level point of view.

The key contributions of this work are (1) proposed 2 practical algorithms RSGM and IRLS, (2) with good theoretical analysis for the landscape and convergence rate. Also, when the outlier ratio exceeds certain ration, experiment results support the superior performance of this work compared with previous DPCP with clear margin [1].

Also limitations of this work are presented in section 6, good to see.

This work did a nice citation of related works from outlier-robust estimation to robust surrogation optimization (some below).

[1] Manolis C Tsakiris and Rene Vidal. Dual principal component pursuit. Journal of Machine Learning Research, 2018.

[2] Zhihui Zhu, Yifan Wang, Daniel Robinson, Daniel Naiman, Rene Vidal, and Manolis Tsakiris. Dual principal component pursuit: Improved analysis and efficient algorithms. In Advances in Neural Information Processing Systems, 2018.

[3] Zhihui Zhu, Tianyu Ding, Daniel Robinson, Manolis Tsakiris, and René Vidal. A linearly convergent method for non-smooth non-convex optimization on the grassmannian with applications to robust subspace and dictionary learning. In Adv. Neural Inf. Process. Syst., 2019.

[4] Ji Zhao. An efficient solution to Non-Minimal case essential matrix estimation. IEEE Trans. Pattern Anal. Mach. Intell., 2022

**Weaknesses:**

From section 5, It is good to see OR2-IRLS outperform the alternative algorithm DPCP [1] when the outlier ratio exceeds certain level, for different number of points to the robust essential matrix estimation problem. However, this can be considered as synthetic data only, and will be nice to see how the proposed algorithm perform on real data such as 3D point cloud road data (e.g., Zhihui Zhu, et al, [2]).

Given the theoretical analysis for both OR2-RSGM and OR2-IRLS are presented in section 4 and Appendix, it seems no discussion or high-level summary about computational cost is given in this work.

The algorithm contribution of OR2-IRLS seems largely based on previous work (e.g., [1-4]) and sort of relative limited.

Maybe I missed, not sure if experimental results in section 5 or Appendix, included any empirical comparison between 2 proposed algorithms in this work?

**Questions:**

Question 1: the experiential results suggest that the performance of estimation algorithm can be improved by taking consideration of manifold constrains. And related to this, not sure if by adding the manifold constrains can lead to the proposed OR2 optimization framework can handle certain tasks that cannot be solve by framework eq. [1] in the paper.

Question 2: in case if not included yet, should be good to see the empirical results comparison between OR2-RSGM and OR2-IRLS.

Question 3: seems there are dual listed reference for Tsakiris & Vidal 2018?

---

### Meta-Review · Area_Chair_j7M7 · 2023-12-03

**Metareview:**

**Summary**

This paper presented a generalized framework to L1 orthogonal regression problem as outlier-robust orthogonal regression over manifolds. The paper designs two algorithms and analyzes their convergence rate. There are experimental results showing that under certain level of noise, the proposed approaches outperform baselines.

**Strengths**

- The proposed algorithms are practical, and the convergence analysis is provided.
- The paper is overall well written and motivated.

**Weaknesses**

- As many reviewers noted, the core idea in algorithmic design and theoretical analysis is largely based on prior works.
- The correctness of main theorems is questionable, as noted by reviewers.
- Some reviewers have pointed out that a number of relevant papers were not discussed or compared.
- Numerical experiments are not comprehensive enough to substantiate the theory.

**Justification For Why Not Higher Score:**

The novelty of the work is limited, due to the large overlap with prior works in terms of algorithm design and analysis. The correctness of the proof is in doubt. Experimental results are not fully convincing.

**Justification For Why Not Lower Score:**

N/A

---

### Decision · Program_Chairs · 2024-01-16

Reject